# The global atmospheric energy transport analysed by a wavelength-based scale separation

**Patrick Johannes Stoll**[1], **Rune Grand Graversen**[2,3], and **Gabriele Messori**[4,5,6,7]

[1]Department of Mathematics and Statistics, Arctic University of Norway, Tromsø, Norway TS1
[2]Department of Physics and Technology, Arctic University of Norway, Tromsø, Norway
[3]Norwegian Meteorological Institute, Tromsø, Norway
[4]Department of Earth Sciences, Uppsala University, Uppsala, Sweden
[5]Centre of Natural Hazards and Disaster Science (CNDS), Uppsala, Sweden
[6]Department of Meteorology, Stockholm University, Stockholm, Sweden
[7]Bolin Centre for Climate Research, Stockholm, Sweden

**Correspondence:** Patrick Johannes Stoll (patrick.stoll@uit.no)

**Abstract.** The atmosphere transports energy polewards by circulation cells and eddies. To the present day, there has been a knowledge gap regarding the preferred spatial scales and physical mechanisms of eddy energy transport. To fill the gap, we separate the meridional atmospheric energy transport in the ERA5 reanalysis by spatial scales and into quasi-stationary and transient flow patterns and latent and dry components.

Baroclinic instability is the major instability mechanism in the transient synoptic scales and is responsible for forming cyclones, anticyclones, and small-scale Rossby waves. At the planetary scales, circulation patterns are often induced by other mechanisms such as flow interaction with orography and land–sea heating contrasts. However, a separation between circulation patterns at the synoptic and planetary scales has yet to be established. We find that both baroclinically induced and transient energy transport is predominantly associated with eddies at wavelengths between 2000 and 8000 km. The maxima in both types of transport occur at wavelengths around 5000 km, in good agreement with linear baroclinic theory. Since these results are independent of latitude, we adapt the scale separation of the energy transport to be based on the wavelength instead of the previously used wavenumber. We define the synoptic transport by the wavelength band between 2000 and 8000 km.

We analyse the annual and seasonal mean in the energy transport components and their inter-annual variability. The scale-separated transport components are fairly similar in both hemispheres. Transport by synoptic waves is the largest contributor to extra-tropical energy and moisture transport, mainly of a transient character, and is influenced little by seasonality. In contrast, transport by planetary waves depends highly on the season and has two distinct characteristics. (1) In the extra-tropical winter, planetary waves are important due to a large transport of dry energy. This planetary transport features the largest inter-annual variability of all components and is mainly quasi-stationary in the Northern Hemisphere but transient in its southern counterpart. (2) In the sub-tropical summer, quasi-stationary planetary waves are the most important transport component, mainly due to moisture transport, presumably associated with monsoons. In contrast to transport by planetary and synoptic waves, only a negligible amount of energy is transported by mesoscale eddies ($< 2000\,\mathrm{km}$).

## 1 Introduction

Atmospheric motions reduce the thermal contrast created by differential solar heating between high and low latitudes (Hadley, 1735). Hence the atmospheric circulation transports large amounts of energy polewards (Oort and Peixóto, 1983) and has thereby a fundamental role in controlling the local weather and climate (e.g. Holton and Hakim,

2013; Vallis, 2017). The energy is primarily transported in the form of warm air (dry energy: comprising mainly enthalpy and potential energy) and water vapour (latent energy) which releases energy when condensating before precipitation (Peixoto and Oort, 1992).

The characteristics of the energy transport differ among latitudinal zones (Trenberth and Stepaniak, 2003a). In the tropics and sub-tropics, where the Coriolis effect is small, energy is predominantly transported by a zonally symmetric meridional overturning circulation, known as the Hadley cell (Hadley, 1735), and monsoon systems organised by quasi-stationary cells (Fig. 1a). In the extra-tropics, eddies transport the majority of the energy further poleward. Atmospheric eddies exist on a large range of scales, from planetary Rossby waves (Rossby, 1939) and transient synoptic Rossby waves, which vertically interact with synoptic cyclones (Bjerknes, 1919), to mesoscale disturbances, such as polar lows (Businger and Reed, 1989). Conventionally, the eddy transport is separated into a quasi-stationary component and a transient component (Fig. 1a), with the former representing monthly-mean eddies and the latter faster-varying deviations from this mean (Oort and Peixóto, 1983). Transient eddies dominate the extra-tropical energy transport in both hemispheres (Fig. 1a), whereas quasi-stationary waves transport a considerable amount of energy in the higher latitudes of the Northern Hemisphere (NH) and the sub-tropics of both hemispheres. However, they are less relevant in the extra-tropical Southern Hemisphere (SH).

Recent studies demonstrate the usefulness of separating the energy transport by spatial scales into planetary and synoptic components. For instance, eddies at these two scales impact the Arctic differently (Baggett and Lee, 2015; Graversen and Burtu, 2016). The scale separation of the meridional energy transport by a zonal Fourier decomposition became popular in recent years. It was applied to study the mechanisms and impacts of energy transport in the Arctic region (Papritz and Dunn-Sigouin, 2020; Graversen et al., 2021; Rydsaa et al., 2021; Hofsteenge et al., 2022) and the NH mid-latitudes (Lembo et al., 2019). Further, numerous studies investigated the atmospheric dynamics in terms of Rossby waves at different spatial scales (e.g. Wirth et al., 2018; Röthlisberger et al., 2019).

So far, the energy transport across all latitudes has only been separated by a fixed wavenumber as shown in Fig. 1b and presented by Graversen and Burtu (2016). The separation between planetary and synoptic scales has often been established between waves CE1 3 and 4 (Baggett and Lee, 2015; Heiskanen et al., 2020). For investigation of the scale of the transport at a specific latitude or a small zonal band, such a separation by wavenumber is appropriate. However, the wavelength associated with a given wavenumber is latitude dependent, which needs to be accounted for when defining the wavenumber separating spatial scales (Heiskanen et al., 2020). For instance, wave 4 corresponds to a wavelength of 8200 km at 35° but only to 2600 km at 75°. These two wave-

lengths can be interpreted to represent different spatial scales. Such a partitioning by wavenumber thus has two caveats: (i) towards the poles, all eddy transport converges to the planetary scale (Fig. 1b). (ii) In the sub-tropics, wavenumbers 1–3 appear to miss parts of planetary transport, as can be inferred from quasi-stationary eddies in the sub-tropical SH (Fig. 1a) being considerably larger than the planetary transport captured by wavenumbers 1–3 (Fig. 1b).

This study proposes a revised partitioning by spatial scales based on wavelength to circumvent the above-mentioned problems associated with separation by wavenumber. With this, we partition the transport into planetary, synoptic, and mesoscale components to understand their role in transporting energy poleward. In addition, we apply the conventional decomposition of the transport into stationary and transient parts (Fig. 1a), as well as dry and latent components (Peixoto and Oort, 1992).

The new scale-separation method is employed to analyse meridional energy transport by the global atmospheric circulation. Previous studies found that transport by quasi-stationary waves is important in the NH but almost negligible in the SH (e.g. Peixoto and Oort, 1992). Such quasi-stationary transport is often associated with the planetary scale, which appears to imply that planetary transport is irrelevant in the SH (e.g. Trenberth and Stepaniak, 2003a). However, this is not the case since planetary waves are strongly represented in the SH as shown in Fig. 1b. In contrast, transport by transient eddies is often associated with baroclinic eddies at the synoptic scale (e.g. Trenberth and Stepaniak, 2003a). However, transport at other scales could be transient as well. In this study, we underscore that the separation of transport into a quasi-stationary and transient contribution differs from a separation into planetary and synoptic scales. However, a correlation exists between the two, especially for the NH.

The following summarises the main research questions posed in this study:

1. At what scales does the atmosphere transport energy, and how does the scale-separation method compare to the conventional separation method?

2. What characterises the atmospheric energy transport in different meridional bands?

These questions are investigated in TS2 Sects. 3 and 4. The main results are then summarised and discussed in Sect. 5. However, first, we present the utilised data and methods.

## 2 Data and methods

### 2.1 ERA5 reanalysis

The atmospheric energy transport for the period 1979–2021 is analysed with ERA5 (Hersbach et al., 2020), the fifth atmospheric reanalysis from the European Centre for Medium-

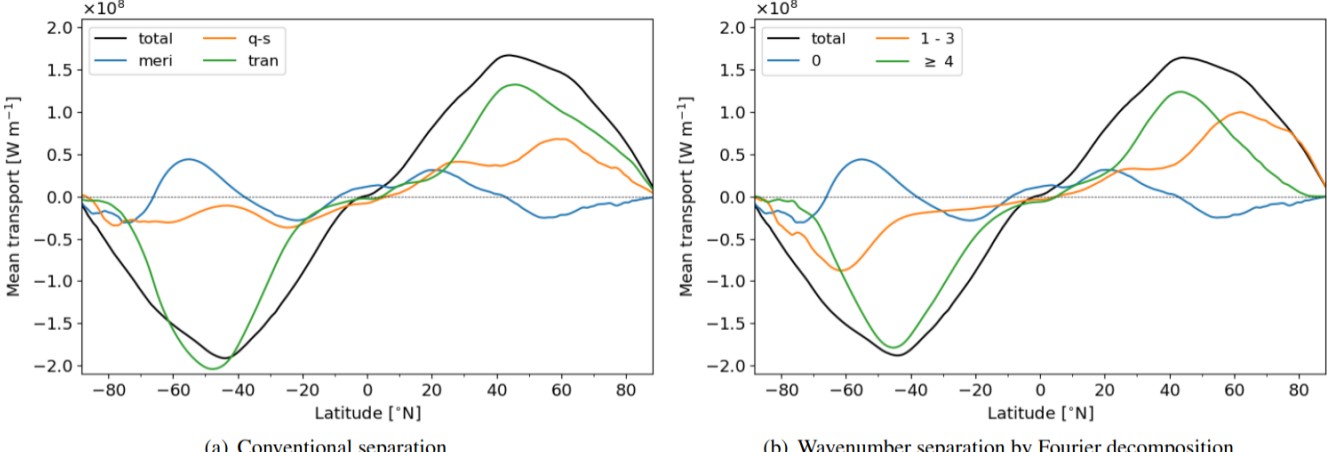

(a) Conventional separation  (b) Wavenumber separation by Fourier decomposition

**Figure 1.** Different separation methods of the vertically integrated, zonal-mean, northward transport of energy from ERA5 as a mean over the years 1979 to 2018. **(a)** The conventional partitioning into the meridional overturning circulation (meri), quasi-stationary (q-s), and transient (tran) eddies, as performed by Oort and Peixóto (1983). **(b)** A decomposition into transport by different spatial scales defined by wavenumbers as suggested by Graversen and Burtu (2016). Transport by wave 0 provides the meridional overturning circulation. The sum of transport by waves 1–3 is often associated with planetary transport, whereas waves with larger wavenumbers are considered to be of synoptic scale.

Range Weather Forecasts (ECMWF). ERA5 provides hourly fields at a spectral truncation of T639, equivalent to a grid spacing of 30 km, and with 137 vertical hybrid-sigma levels. For this study, fields of air temperature, specific humidity, geopotential height and horizontal wind components at all hybrid levels, and surface pressure and surface topography are used on a $0.25° \times 0.25°$ horizontal grid spacing. Since reanalysis data are prone to include mass-flux inconsistencies (Trenberth, 1991), a barotropic mass-flux correction is applied to the wind field before calculating the energy transport (Graversen, 2006).

## 2.2 Zonal mean versus zonal integral

The zonal integral of the ERA5 energy transport (Fig. S1a) confirms that found in previous studies using datasets with lower spatial resolution (Peixoto and Oort, 1992; Trenberth and Caron, 2001; Graversen and Burtu, 2016). For instance, the zonally integrated poleward transport peaks at $4.8 \times 10^{15}$ W in the NH and $5.6 \times 10^{15}$ W in the SH at a latitude of 41° in both hemispheres. By computing the zonal integral of the energy transport, which depends on the length of the longitude circle, the transport becomes small at high latitudes since the longitudes converge (Fig. S1a). However, the local transport, expressed by the zonal mean, is considerable also in the polar regions (Fig. S1b). Hence, to compare the local importance of the atmospheric energy transport across all latitudes, we take a zonal-mean perspective which provides the transport through an atmospheric column with 1 m width. The latitude of maximum zonal-mean transport, namely the location at which the largest local meridional transport occurs, is at a latitude of 45° in both hemispheres

(Fig. S1b), different from the maximum zonally integrated transport peaking at 41°.

For the calculation of convergence of energy transport, a 2° running-mean filter is used along the meridional dimension before computing the meridional derivatives.

## 2.3 Decomposition of the energy transport

The atmospheric energy transport and its components are characterised by a large day-to-day variability (Swanson and Pierrehumbert, 1997; Messori and Czaja, 2013). Here we focus on the annual-mean and seasonal-mean energy transport, firstly to investigate the time-mean behaviour of the atmospheric circulation and secondly to compare the newly introduced separation of the eddy transport based on spatial scales – described below – with the conventional separation into quasi-stationary and transient transport introduced by Oort and Peixóto (1983) and commonly used in the literature (e.g. Trenberth and Stepaniak, 2003a, b; Kaspi and Schneider, 2013). The conventional separation obtains quasi-stationary eddies from monthly-mean fields; hence a comparison is only possible from a time-mean perspective. Quasi-stationary eddies could be derived differently, for example, by application of a temporal low-pass filter. This would enable the comparison of the separation method by scale and by quasi-stationary and transient components without taking time averages. However, it would prevent a direct comparison with the literature and require a new ad hoc analysis, which is beyond the scope of this study.

We define the vertically integrated, zonal-mean, time-mean, meridional transport of energy as

$$\widetilde{vE} = \int_0^{p_s} [\overline{vE}] \frac{\mathrm{d}p}{g},\qquad(1)$$

where $v$ is the meridional wind, $E$ is the atmospheric energy, $p$ is pressure, $p_s$ is surface pressure, $g$ is the gravitational constant, $\overline{\cdot}$ denotes a monthly time average, and $[\cdot]$ is CE2 a zonal average. We decompose the total atmospheric energy transport, $\widetilde{vE}$, in three ways that can be applied in succession:

Firstly, CE3 the total atmospheric energy transport, $\widetilde{vE}$, is partitioned into latent energy transport, $\widetilde{vQ}$, and dry energy transport, $\widetilde{vD}$, following Oort and Peixóto (1983). This is achieved by separating the total atmospheric energy, $E$, in Eq. (1) into its latent component, $Q$, and dry component, $D$, which comprises the enthalpy, $c_p T$; potential energy, $gz$; and kinetic energy, $\frac{v^2}{2}$, as

$$E = Q + D = Lq + \left(c_p T + gz + \frac{\boldsymbol{v}^2}{2}\right),\qquad(2)$$

with $q$ being specific humidity, $L$ being the latent heat release by condensation, $c_p$ being specific heat capacity at constant pressure, $T$ being temperature, and $\boldsymbol{v}$ being the horizontal wind vector.

Secondly, the total energy transport, $\widetilde{vE}$, expressing a time-mean transport, is separated into transport by transient eddies, $\widetilde{vE}^{\mathrm{tr}} = \int_0^{p_s} [\overline{v'E'}] \frac{\mathrm{d}p}{g}$; quasi-stationary eddies, $\widetilde{vE}^{\mathrm{q\text{-}s}} = \int_0^{p_s} [\overline{v}^* \overline{E}^*] \frac{\mathrm{d}p}{g}$; and the mean meridional circulation, $\widetilde{vE}^{\mathrm{md}} = \int_0^{p_s} [\overline{v}][\overline{E}] \frac{\mathrm{d}p}{g}$, by partitioning the zonal-mean, time-mean transport, $[\overline{vE}]$, in Eq. (1) according to Oort and Peixóto (1983) and Peixoto and Oort (1992) by

$$\begin{aligned}[\overline{vE}] &= [\overline{v}\,\overline{E}] + [\overline{v'E'}] \\ &= [\overline{v}][\overline{E}] + [\overline{v}^* \overline{E}^*] + [\overline{v'E'}],\end{aligned}\qquad(3)$$

with $\cdot'$ denoting anomalies from the time mean and $\cdot^*$ denoting zonal anomalies from the zonal mean. The first equality separates the time-mean transport into the transport by the time-mean fields, $[\overline{vE}]$, and by faster-varying fluctuations, $[\overline{v'E'}]$. The second equality partitions the time-mean term into a zonally symmetric transport component, $[\overline{v}][\overline{E}]$, and a contribution by zonal waves, $[\overline{v}^* \overline{E}^*]$. The duration of the time mean, $\overline{\cdot}$, for which eddies are considered quasi-stationary, is arbitrary. However, conventionally the mean is taken over 1-month periods (e.g. Oort and Peixóto, 1983). Hereby, the quasi-stationary eddy transport, $[\overline{v}^* \overline{E}^*]$, provides the transport given by the monthly-mean eddies, and the transient transport, $[\overline{v'E'}]$, is the faster varying component. The annual-mean energy transport partitioned in this conventional manner is depicted in Fig. 1a. The transport of latent energy, $\widetilde{vQ}$, and dry energy, $\widetilde{vD}$, is partitioned similarly.

Thirdly, the total energy transport, $\widetilde{vE}$, is separated into transport by eddies at the planetary scale, $\widetilde{vE}_{\mathrm{p}}$; the synoptic scale, $\widetilde{vE}_{\mathrm{sy}}$; and the mesoscale, $\widetilde{vE}_{\mathrm{ms}}$, as well as the meridional circulation, $\widetilde{vE}_{\mathrm{md}}$. For the partitioning into spatial scales, a Fourier decomposition is applied along each latitude, $\phi$; at each vertical model level; and at every time step for $E$ and $v$[1] following Graversen and Burtu (2016). The Fourier series of $E$ yields

$$E = \frac{a_0^E}{2} + \sum_{n=1}^{\infty} a_n^E \cos\left(\frac{n2\pi x}{d}\right) + b_n^E \sin\left(\frac{n2\pi x}{d}\right),\qquad(4)$$

where $x$ is the zonal coordinate, $d = 2\pi a \cos(\phi)$; $a = 6371\,\mathrm{km}$ (Earth's radius); and the Fourier coefficients, $a_n^E$ and $b_n^E$, are given as

$$\begin{aligned}a_n^E &= \frac{2}{d} \oint E \cos\left(\frac{n2\pi x}{d}\right) \mathrm{d}x, \\ b_n^E &= \frac{2}{d} \oint E \sin\left(\frac{n2\pi x}{d}\right) \mathrm{d}x,\end{aligned}\qquad(5)$$

where $n$ is zonal wavenumber. The Fourier coefficients for the meridional wind, $v$, $a_n^v$ and $b_n^v$, are derived in a similar manner.

Based on this Fourier decomposition, the time-mean, zonal-mean energy transport can be written as

$$[\overline{vE}] = \frac{1}{4}\overline{a_0^v a_0^E} + \frac{1}{2}\sum_{n=1}^{\infty} \overline{\{a_n^v a_n^E + b_n^v b_n^E\}}.\qquad(6)$$

Note that the cross terms $a_n^v$ and $a_m^E$ with $n \neq m$, and similarly $b_n^v$ and $b_m^E$, vanish when computing a zonal average since the Fourier coefficients feature an orthogonal basis. In the following, we show how this decomposition relates to the decomposition by Oort and Peixóto (1983) expressed in Eq. (3).

The first term in Eq. (6) can be approximated by the first term on the right-hand side (RHS) of Eq. (3):

$$\frac{1}{4}\overline{a_0^v a_0^E} = \overline{[v][E]} = [\overline{v}][\overline{E}] + \overline{[v'][E']} \simeq [\overline{v}][\overline{E}]\qquad(7)$$

since $\overline{[v'][E']}$, the time mean of the transient part of the meridional circulation, is small. Hence, the time-mean transport by the meridional overturning circulation can also be written as $\widetilde{vE}_{\mathrm{md}} = \int_0^{p_s} \frac{1}{4}\overline{a_0^v a_0^E} \frac{\mathrm{d}p}{g}$.

The second term in Eq. (6) is a time-mean transport by eddies, $[\overline{v^* E^*}]$, and can be separated into stationary and transient eddy transport:

$$\frac{1}{2}\sum_{n=1}^{\infty} \overline{\{a_n^v a_n^E + b_n^v b_n^E\}} = [\overline{v^* E^*}] = [\overline{v}^* \overline{E}^*] + [\overline{v'^* E'^*}].\qquad(8)$$

---

[1]To be precise, the Fourier decomposition is applied on the mass flux, $v\,\mathrm{d}p/g$, since the thickness, $\mathrm{d}z$, times density, $\rho$, providing the mass thickness of the model layers, $\mathrm{d}p/g = -\rho\,\mathrm{d}z$, is non-constant in hybrid-sigma coordinates.

The stationary eddy part is the second term on the RHS of Eq. (3) and can also be expressed by

$$[\overline{v^* E^*}] = \frac{1}{2} \sum_{n+1}^{\infty} \left\{ \overline{a}_n^v \overline{a}_n^E + \overline{b}_n^v \overline{b}_n^E \right\}. \tag{9}$$

The transient eddy part, $[\overline{v'^* E'^*}]$, is the difference between the total transient transport, $[\overline{v' E'}]$ (term 3 on RHS of Eq. 3), and the transient part of the meridional circulation $\overline{[v'][E']}$, which is again small in the time mean, hence

$$[\overline{v'^* E'^*}] = [\overline{v' E'}] - \overline{[v'][E']} \simeq [\overline{v' E'}]. \tag{10}$$

Having shown the relation of the terms in the Fourier decomposition with the transient and quasi-stationary terms in the conventional decomposition, we now introduce the scale separation of the eddy transport. The instantaneous zonal-mean transport by wave $n$ is given by

$$[v^* E^*]_n = \frac{1}{2} \left( a_n^v a_n^E + b_n^v b_n^E \right). \tag{11}$$

To separate the energy transport between two scales based on a predefined wavelength, $\lambda_{\text{sep}}$, the latitude-dependent ($\phi$) wavenumber of separation, $n_{\text{sep}}$, is computed:

$$n_{\text{sep}} = \frac{2\pi \cdot a \cdot \cos(\phi)}{\lambda_{\text{sep}}}. \tag{12}$$

The wavenumber of separation between planetary and synoptic eddies, $n_{\text{p/s}}$, is computed at each latitude from the wavelength $\lambda_{\text{p/s}} = 8000$ km (lower black solid line in Fig. 2). Synoptic and mesoscale eddies are separated at a wavelength of $\lambda_{\text{s/m}} = 2000$ km. In Sect. 3, we argue for the usage of these two wavelengths for scale-separating the energy transport. The wavenumbers of separation, as defined here, are real numbers. To ensure a continuous separation (Fig. S2b), instead of abrupt transitions (Fig. S2a), the transport by planetary, synoptic, and mesoscale eddies is defined by

$$[v^* E^*]_p = \sum_{n=1}^{\lfloor n_{\text{p/s}} \rfloor} [v^* E^*]_n$$
$$+ \left( n_{\text{p/s}} - \lfloor n_{\text{p/s}} \rfloor \right) \cdot [v^* E^*]_{\lceil n_{\text{p/s}} \rceil}, \tag{13}$$

$$[v^* E^*]_{\text{sy}} = \left( \lceil n_{\text{p/s}} \rceil - n_{\text{p/s}} \right) \cdot [v^* E^*]_{\lceil n_{\text{p/s}} \rceil}$$
$$+ \sum_{n=\lceil n_{\text{p/s}} \rceil + 1}^{\lfloor n_{\text{s/m}} \rfloor} [v^* E^*]_n$$
$$+ \left( n_{\text{s/m}} - \lfloor n_{\text{s/m}} \rfloor \right) \cdot [v^* E^*]_{\lceil n_{\text{s/m}} \rceil}, \tag{14}$$

$$[v^* E^*]_{\text{ms}} = \left( \lceil n_{\text{s/m}} \rceil - n_{\text{s/m}} \right) \cdot [v^* E^*]_{\lceil n_{\text{s/m}} \rceil}$$
$$+ \sum_{n=\lceil n_{\text{s/m}} \rceil + 1}^{\infty} [v^* E^*]_n, \tag{15}$$

with $\lfloor \cdot \rfloor$ denoting rounding to the integer part and $\lceil \cdot \rceil$ rounding to the least integer. This provides the latitude-dependent

separation depicted in Fig. 2 between the black curves at 2000 and 8000 km.

The partitioning of Eqs. (13)–(15) is applied to calculate the time-mean transport by planetary eddies, $\widetilde{vE}_{\text{p}} = \int_0^{p_s} [\overline{v^* E^*}]_{\text{p}} \frac{dp}{g}$; synoptic eddies, $\widetilde{vE}_{\text{sy}} = \int_0^{p_s} [\overline{v^* E^*}]_{\text{sy}} \frac{dp}{g}$; and mesoscale eddies, $\widetilde{vE}_{\text{ms}} = \int_0^{p_s} [\overline{v^* E^*}]_{\text{ms}} \frac{dp}{g}$. It is in a similar manner applied to calculate the transport by quasi-stationary eddies at different scales, for instance the transport by quasi-stationary planetary eddies, $\widetilde{vE}_{\text{p}}^{\text{q-s}} = \int_0^{p_s} [\overline{v^* E^*}]_{\text{p}} \frac{dp}{g}$. The transport by transient eddies at the different scales is simply obtained from the difference between the time-mean transport and the quasi-stationary transport. For instance, the transport by transient planetary eddies is calculated as $\widetilde{vE}_{\text{p}}^{\text{tr}} = \widetilde{vE}_{\text{p}} - \widetilde{vE}_{\text{p}}^{\text{q-s}}$.

An underlying interpretation is that at a longitude circle with for example 20 000 km extent (at a latitude of about 60°), wave 1 includes the transport at a scale of and larger than the longitude cycle, wave 2 is between 10 000 and 20 000 km, and so forth. Hereby, wave 3 with a wavelength of 6700 km provides the transport between its wavelength and 10 000 km and is therefore partitioned between the synoptic and planetary scales. Since the wavenumber of separation between the planetary and synoptic scale is $n_{\text{p/s}} = 2.5$ at a latitude of 60°, this wave is equally partitioned between the two scales (second term in Eq. 13 and first term in Eq. 14).

This continuous partitioning of the transport by the first wave smaller than 8000 km leads to the synoptic scale including less transport than the application of a strict separation (Fig. S2). Hence, waves at a wavelength around and larger than 8000 km are associated with the planetary scale, whereas the synoptic scale constitutes waves strictly smaller than 8000 km. Spectra depicting the separation into the scale components at different latitudes are provided in Fig. S3, and a detailed illustration of the separation is provided in Sect. S3 in the Supplement. It should be noted that Wolf and Wirth (2017) also applied a continuous scale separation for the latitude-dependent detection of Rossby wave packages. The same procedure of scale separation is applied to the latent and dry energy transport.

Generally, the scale separation based on the Fourier decomposition is non-local, implying that the whole latitude circle influences the obtained eddies (Heiskanen et al., 2020). Therefore the Fourier decomposition is useful if the transport across the circle is governed by similar eddy scales, which we can observe from meteorological weather maps along zonal bands. Arguably, the zonal and meridional scales of atmospheric eddies match: from the investigation of meteorological weather maps, we know (i) that synoptic-scale cyclones are circular to the first order and (ii) that the meridional extent, i.e. the amplitude, of Rossby waves, appears to roughly match the distance between a trough and a ridge in the zonal direction. Hence, as many other studies (e.g. Graversen and Burtu, 2016; Lembo et al., 2019), we interpret

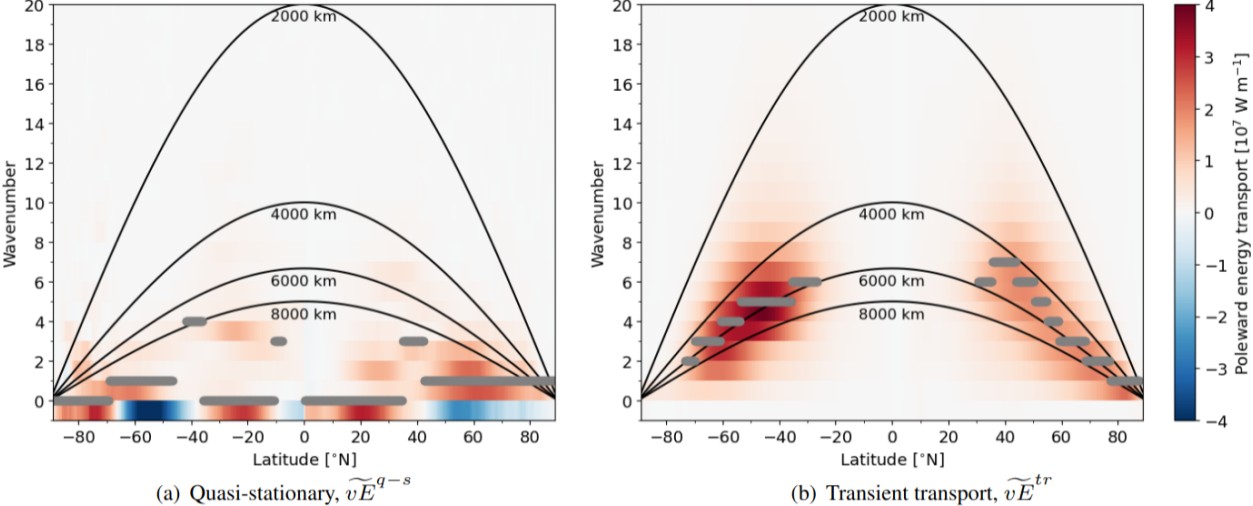

**Figure 2.** The annual-mean, zonal-mean Fourier-decomposed poleward transport of energy from ERA5 by each wave at all latitudes. Decomposition of the **(a)** quasi-stationary and **(b)** transient energy transport. The wavenumbers corresponding to wavelengths of 2000, 4000, 6000, and 8000 km are depicted by black curves. The first and last curve separate the transport into the mesoscale, synoptic scale, and planetary scale. At each latitude the wave of maximal poleward energy transport is displayed with a grey dot. The dot is not displayed at latitudes where this wave is responsible for less than 5 % of the total quasi-stationary/transient transport of the latitude with maximum transport.

the zonal wavenumber of an eddy, which is associated with a zonal wavelength at a given latitude, as its spatial scale.

## 3   Wavelengths utilised for scale separation

Orlanski (1975) defines the mesoscale to be smaller than 2000 km, which is commonly accepted and hence used in this study as the lower threshold for the synoptic scale. However, a widely agreed separation between the planetary and synoptic scales is not established. Below, we argue for placing the separation between the planetary and synoptic scales at a wavelength of 8000 km, based on the argument that baroclinic instability is the major instability and hence the physical mechanism of wave generation at the synoptic scale. The exact value is to some degree arbitrary. As previously presented, this partitioning is done continuously such that transport at and around the wavelength of 8000 km is associated with the planetary scale, whereas synoptic waves are strictly smaller than this threshold.

### 3.1   Theoretical baroclinic argument

Baroclinic instability is recognised as the single dominant instability of the synoptic scale (Markowski and Richardson, 2011, p. 4). Thus, the synoptic scale should include most of the energy transports associated with eddies developing by baroclinic instability. Synoptic eddies are typically identified as cyclones and anticyclones in sea-level pressure, as well as shorter Rossby waves (Frederiksen, 1978) often referred to as transient synoptic Rossby waves (e.g. Röthlisberger et al., 2019; Ali et al., 2021). The cyclones and anticyclones in-

teract vertically and baroclinically with the synoptic Rossby waves, generally recognised by upper-tropospheric oscillations of the jet stream. The theoretical scale (wavelength) of baroclinic eddies is given by 3.9 times the Rossby deformation radius, $L_d$, and hence estimated to be 4000 km by Vallis (2017, p. 354) and 4800 km by Stoll et al. (2021) using more exact values obtained from extra-tropical cyclones. Note that a low-pressure (high-pressure) system spans half a wavelength and has accordingly a typical diameter of around 2000 km.

A wavelength band between 2000–8000 km appears appropriate to capture the majority of the transport associated with baroclinically induced synoptic eddies for three reasons. (i) Cyclones and anticyclones feature some variability in their size but with a typical diameter between 1000 and smaller than 4000 km. (ii) Short, synoptic Rossby waves are considered to occur at zonal wavelengths within this band, with meridional amplitudes of around half a wavelength, as further discussed in Sect. 3.4. (iii) The non-local Fourier decomposition of the energy transport in situations of localised synoptic cyclones captures considerable transport at neighbouring waves to the cyclone (Heiskanen et al., 2020, Fig. 3), which may be considered a weakness of the Fourier decomposition. However, the band of wavenumbers around the localised eddy appears appropriate to capture the eddy.

### 3.2   Statistical baroclinic argument

Figure 3 depicts the anomalous daily energy transport by different waves at a lag time of 4 d after enhanced meridional temperature gradients (90th percentile). Different lag times ($> 0$) provide similar results (Fig. S4), and the pre-

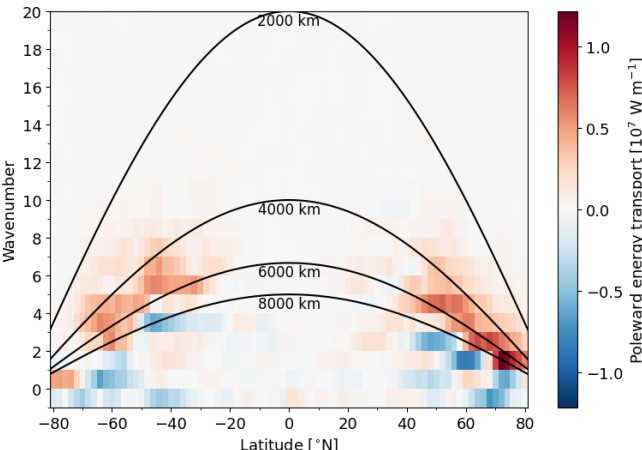

**Figure 3.** The composite of the anomalous poleward energy transport by individual waves ($[vE]_n$ of Eq. 11) at different latitudes 4 d after situations of anomalous high (90th percentile) meridional temperature contrasts around the latitude of energy transport. The anomalies on a daily resolution are computed from the 21 d and 9-year running-mean climatology. The meridional temperature contrast is computed by the difference between the zone 10° equatorward and 10° poleward of the latitude of interest. Hence, red colours denote waves transporting more energy polewards 4 d after situations of enhanced meridional temperature differences. The wavenumbers corresponding to wavelengths of 2000, 4000, 6000, and 8000 km are depicted by black lines. Wave 0 is corrected by the meridional mass flux to express the instantaneous zonal-mean transport according to Lembo et al. (2019).

sented one of 4 d is in good agreement with Fig. 1a of Baggett and Lee (2015). The separation between waves featuring enhanced and decreased transport as a response to the increased thermal contrast is at around 8000 km for all latitudes in the extra-tropics. Waves whose energy transport is most closely associated with the increased temperature gradient have a wavelength between 3000 and 8000 km with a maximum transport anomaly for wavelengths around 5000–6000 km. Since baroclinic instability is induced by a horizontal temperature gradient, this provides strong statistical evidence that baroclinic instability is an important development mechanism for waves at wavelengths between 2000 and 8000 km, which is here defined as the synoptic scale. In the following, we refer to the anomalous energy transport 4 d after situations of enhanced meridional temperature gradients as a "baroclinically induced transport anomaly". The strength of the baroclinically induced transport anomaly by the synoptic waves (Fig. 3 at a wavelength between 2000 and 8000 km) is around a third of the time-mean energy transport of these waves (Fig. 2). The mean meridional temperature contrast will also induce baroclinic waves transporting energy, which is not captured by this methodology.

The latitudinal independence of the wavelength of the baroclinically induced transport anomaly provides a major argument for using a wavelength threshold to separate between the synoptic and planetary scales rather than a wavenumber threshold. It is surprising that the scale of maximum baroclinically induced transport anomaly is independent of the latitude since the Rossby deformation radius, $L_d = \frac{NH}{f}$, estimating the size of baroclinic eddies (Vallis, 2017), depends inversely on the Coriolis parameter, $f$, increasing with latitude and depends linearly on the depth of the troposphere, $H$, and the tropospheric static stability, $N$, which rather decrease with latitude (Stoll et al., 2021). Hence, baroclinic eddies would be expected to be smaller at higher latitudes. A hypothesis for the latitude independence of the baroclinic eddies is as follows. Most extra-tropical cyclones originate from the mid-latitudes, where the meridional temperature contrast is the largest. The size of a cyclone is set during the genesis stage when the fastest-growing mode is prevailing. Many cyclones propagate to higher latitudes along the diagonal axis of the storm tracks (Shaw et al., 2016) and may preserve their original size.

We note that processes other than baroclinic instability may contribute to the formation of the here-defined synoptic eddies, such as heating contrasts. A future study will further investigate the drivers of eddy transport at different scales.

### 3.3 Transient and quasi-stationary eddies

In the following, we show that the partitioning at a wavelength of 8000 km approximately captures our intuitive understanding that synoptic cyclones and synoptic Rossby wave packets are transient by nature, whereas most quasi-stationary eddies are at planetary scales since they may be constrained by large-scale orography and semi-stationary thermal forcing, such as heating contrasts between ocean and land (Vallis, 2017).

Indeed, the spectral decomposition of the annual-mean energy transport, $\widetilde{vE}$, at different latitudes reveals that most eddies smaller than 8000 km are of a transient nature, whereas most of the quasi-stationary transport is at scales larger than 8000 km (Fig. 2). This is in good agreement with Dell'Aquila et al. (2005), who found in the average of the extra-tropical NH, that wave 3, here defined as planetary scale, features a typical period of around a month, whereas smaller eddies, here associated with the synoptic scale, are characterised by weekly or daily periods.

For the transient energy transport, the wavenumber of maximum transport is 6 or 7 in the sub-tropics and decreases towards the poles such that the corresponding wavelength of maximum transient transport is around 5000 to 6000 km for all latitudes. Also in the moisture transport, the transient component reaches its maximum around 5000 km (Fig. S5), which is in good agreement with Lee et al. (2019). This wavelength corresponds well to the preferred scale of baroclinic eddies (4000–5000 km: Vallis, 2017; Stoll et al., 2021) and of baroclinically induced transport anomalies presented in the previous section. The latitudinal independence of the wavelength of maximum transient transport supports sepa-

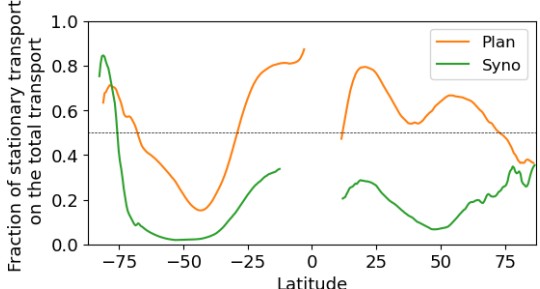

**Figure 4.** The fraction of the quasi-stationary part of the transport by planetary and synoptic waves as a function of latitude in ERA5.

rating the planetary and synoptic scale by using wavelength instead of wavenumber.

The preferred wavenumber and wavelength of the quasi-stationary transport, $\widetilde{vE}^{\text{q-s}}$, are latitude dependent (Fig. 2a). The wavelength of maximum quasi-stationary transport is larger than 8000 km everywhere besides close to the North Pole where the Fourier decomposition is not applicable and around 40° S where the quasi-stationary transport is almost negligible. A short analysis of the preferred scales of the quasi-stationary transport is provided in Sect. S5.

The separation of synoptic and planetary waves at a wavelength of 8000 km captures our intuitive understanding that (i) most transient transport occurs at the synoptic scale, whereas (ii) most quasi-stationary transport is at the planetary scale. The inverse is valid for the former but not the latter: planetary waves at a wavelength larger than 8000 km can be both transient and quasi-stationary (Fig. 2). In the latitude band between 70 and 30° S and poleward of 70° N, zones mainly characterised by open ocean, the transient component within the planetary transport is larger than the quasi-stationary component (Fig. 4). At latitudes with more land, the quasi-stationary component is responsible for 50 %–80 % of the planetary transport, meaning that also at these latitudes a considerable proportion of the planetary transport is transient. Also the moisture transport, $\widetilde{vQ}$, comprises a planetary component that features both transient and quasi-stationary contributions, similar to the total energy transport, $\widetilde{vE}$ (Fig. S6).

In contrast to planetary transport, the inverse of (i) above, that synoptic transport is mainly (70 %–100 %) of a transient nature, is valid at almost all latitudes (Fig. 4). This coincides with the transient character of synoptic cyclones and Rossby waves of short wavelengths. The small quasi-stationary contribution (0 %–30 %) to the synoptic transport is attributed to preferred spatial locations for synoptic activity. For instance, the NH Atlantic sector features more cyclonic activity than other longitudes, resulting in increased quasi-stationary transport in the time mean. This can be inferred from Rydsaa et al. (2021), who show a large time-mean synoptic transport in the Atlantic sector for strong latent transport events in winter at 70° N. However, in a zonal-mean perspective,

the quasi-stationary contribution to the synoptic transport is small ($< 30\%$) compared to its transient part. Hence, for simplicity's sake, the synoptic transport is not separated into a transient and quasi-stationary contribution in the remainder of this study. An exception for the transient character of synoptic transport is over Antarctica, which is characterised by katabatic flows advecting cold air towards lower latitudes, occurring at preferred locations of drainage, hence including a large stationary component.

We conclude that the conventional decomposition of the transport into stationary and transient parts (Oort and Peixóto, 1983) is to some degree related but not equivalent to a separation into planetary and synoptic waves.

## 3.4 Comparison to previous studies

Previous studies performing a wave decomposition qualitatively agree on the separation at a wavelength of 8000 km:

Firstly, CE4 Baggett and Lee (2015) perform a Fourier decomposition of energy characteristics for the entire NH and demonstrate different life cycle behaviour of long (planetary) and short (synoptic) waves for a separation between waves 3 and 4. Ali et al. (2021) identify recurrent synoptic-scale transient Rossby wave packets in the mid-latitudes at wavenumbers between 4 and 15. Wirth et al. (2018) note that planetary-scale Rossby waves are typically characterised by zonal wavenumbers 1–3, whereas higher wavenumbers characterise synoptic-scale Rossby waves.

Secondly, their separation into planetary and synoptic eddies by wavenumber is similar to the here-applied partitioning by wavelength at a CE5 latitude of 53°, which is close to the location of maximum eddy activity (Fig. 1). For a hemisphere-wide separation into planetary and synoptic eddies with a single wavenumber threshold as utilised by Baggett and Lee (2015) and Ali et al. (2021), it appears crucial to capture the partitioning around this latitude.

Thirdly, at 70° N, a wavelength of 8000 km is associated with a separation number $n_{\text{sy}} = 1.7$, meaning that 70 % of wave 2 is associated with the planetary scale and 30 % is associated with the synoptic scale. Accordingly, the lead–lag regression of the Arctic temperature on the latent transport presented in Fig. 6 of Graversen and Burtu (2016) shows a clear difference between wave 1, leading to heating of the Arctic, and waves 3 and larger, having a baroclinic signal. Wave 2 appears to share characteristics of both wave groups, in agreement with the here-applied partitioning of wave 2 at 70° N. Also Heiskanen et al. (2020) find a good attribution of the meridional energy transport by an idealised cyclone to the synoptic scale if the separation between the planetary and synoptic scales is performed at rather small wavenumbers, even though they do not test a separation between waves 2 and 3.

In conclusion, separating at wavelengths of 8000 km is consistent with previous interpretations of planetary-scale and synoptic-scale transport. However, as mentioned earlier,

**Table 1.** The applied terms to describe latitudinal bands in this study.

| Climate zone | Latitude band |
|---|---|
| Equatorial region | $< 10°$ |
| Tropics | $< 23°$ |
| Sub-tropics | $23–35°$ |
| Extra-tropics | $35–90°$ |
| Mid-latitudes | $35–60°$ |
| Polar boundary | Around $60°$ |
| Polar regions | $60–90°$ |

a sharp threshold likely does not exist. Therefore, the separation wavelength of 8000 km is compared to wavelengths of 10 000 and 6000 km in the Supplement (Fig. S7). The main results of this study are not affected by the exact choice of the separation wavelength, as shortly discussed in the Supplement.

## 4 Organisation of the global energy transport

In this section, we analyse the atmospheric energy transport from ERA5 by utilising the scale-separation method. The applied latitudinal bands used in this study are provided in Table 1.

### 4.1 Annual-mean transport

#### 4.1.1 Overview

The meridional atmospheric energy transport features considerable similarities in the two hemispheres in most components and for most climate zones. Hence, to simplify the comparison of the two hemispheres, we display the poleward transport of both hemispheres at different latitudes on a common $x$ axis (Fig. 5a–c).

The annual-mean, zonal-mean poleward energy transport, $\widetilde{vE}$, is seamless in the sense that it resembles half of a sine curve for both hemispheres (black lines in Fig. 5a) as noted by Trenberth and Stepaniak (2003b). However, in the sub-tropics and mid-latitudes, the energy transport is $\sim 15\%$ larger in the SH than in the NH. The difference is approximately balanced by more oceanic transport in the NH (Trenberth and Caron, 2001).

The energy transport, $\widetilde{vE}$, leads to an annual-mean divergence of around $40\,\mathrm{W\,m^{-2}}$ in the tropics and sub-tropics and convergence poleward of a latitude of $40°$ of up to $100\,\mathrm{W\,m^{-2}}$ in the polar regions (Fig. 5d) in good agreement with Trenberth and Stepaniak (2003a). The smaller divergence close to the Equator is due to oceanic currents transporting heat to the sub-tropics, creating a cold tongue of sea-surface temperatures over the equatorial Pacific (Trenberth and Stepaniak, 2003b).

Despite the total energy transport, $\widetilde{vE}$, appearing seamless, different transport mechanisms are important in the different climate zones, as reflected by considerable variations across latitudes in the moisture and dry transports (Fig. 5b and c) and the scale components of the transport.

The total annual-mean moisture transport, $\widetilde{vQ}$, of both hemispheres features equatorward extremes at around $10°$ (Fig. 5b), poleward maxima at around $40°$, and decaying tails towards the poles. This leads to moisture divergence in the non-equatorial tropics and sub-tropics, whereas moisture converges in the equatorial regions and extra-tropics (Fig. 5e). The moisture transport is generally stronger in the Southern Hemisphere than Northern Hemisphere due to more evaporation on the water surfaces of the SH. Further, some moisture is transported across the Equator from the SH to the NH, leading to the highest convergence of moisture at around $7°\,\mathrm{N}$, which is in the annual mean the approximate location of the intertropical convergence zone (ITCZ).

The dry transport, $\widetilde{vD}$, is plateau-like between a latitude of $10–65°$ (Fig. 5c) and decays towards the Equator and poles. Hence, it is mainly responsible for divergence in the equatorial regions and convergence in the polar regions (Fig. 5f). However, the scale components have different roles in the different climate zones.

#### 4.1.2 Synoptic transport

Synoptic-scale waves dominate the energy transport in the mid-latitudes of both hemispheres (Fig. 5a). However, synoptic transport is around $40\%$ stronger in the SH than in the NH. This is in broad agreement with the finding of Peixoto and Oort (1992) that transient transport is more relevant in the SH, but it should be kept in mind that some of the transient transport occurs at the planetary scale (Fig. 4). Despite a broad spectrum of possible waves, most energy is transported by eddies at wavelengths between 2000 and 8000 km, demonstrating the dominant role of synoptic waves for the mid-latitudinal energy transport (Fig. S7). This provides a different perspective from previous studies finding that transient eddies are responsible for the majority of mid-latitudinal energy transport (e.g. Peixoto and Oort, 1992), since planetary waves may also be transient. The peak transport by synoptic waves is at a latitude of around $45°$, the location of the climatological storm tracks (Hoskins and Hodges, 2002, 2005), and coincides with the zone of largest total energy transport. Hence, synoptic waves transport energy from the sub-tropics ($< 40°$) to the extra-tropics (Fig. 5d).

Extra-tropical synoptic waves transport approximately two-thirds of their energy in dry form and one-third in latent form (Fig. 5b, c). The latent contribution is slightly higher in the mid-latitudes close to the sub-tropics and lower in the polar regions, where the cold atmosphere cannot hold much water vapour. Despite transporting more dry than moist energy, synoptic eddies are responsible for most extra-tropical moisture transport. Especially in the polar regions, almost all

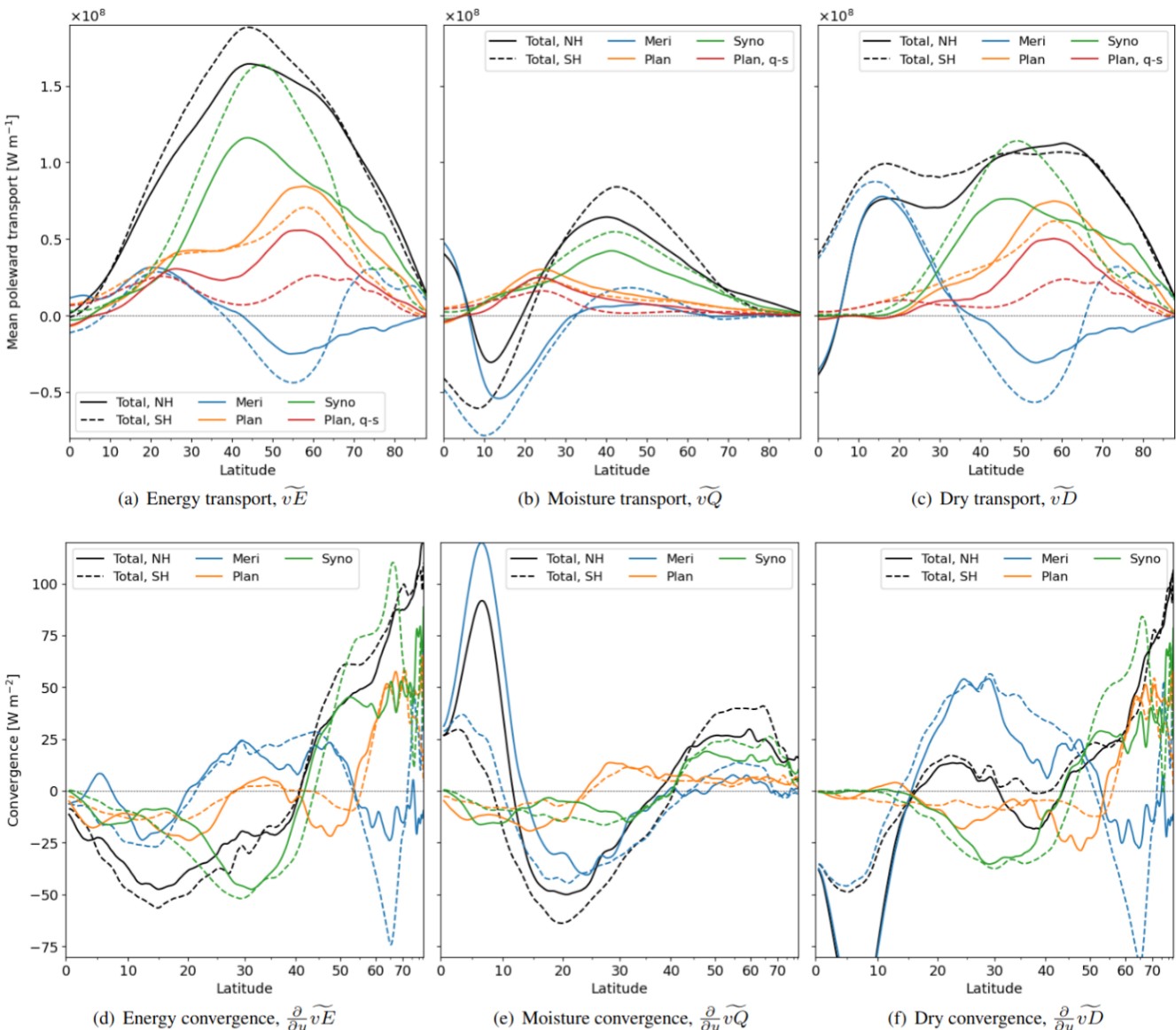

**Figure 5. (a)** Annual-mean, zonal-mean poleward atmospheric energy transport from ERA5. The transport of the Northern Hemisphere and Southern Hemisphere are depicted in solid and dashed lines, respectively. The total energy transport (black) is divided into the zonally symmetric meridional overturning circulation (blue) and wave components at the planetary (orange, waves $\geq 8000$ km) and synoptic scales (green, $< 8000$ km). The quasi-stationary contribution of the planetary transport on a monthly scale is depicted in red. **(b, c)** As panel **(a)** but for the latent and dry energy transport components, respectively. **(d–f)** Resulting convergence of atmospheric energy from the transport components presented in panels **(a)–(c)** but for simplicity omitting the convergence of quasi-stationary planetary transport. The latitudes on the $x$ axis are scaled to represent equal surface areas such that the integrated convergence in each component becomes zero.

moisture transport is by synoptic waves, whereas dry transport in the polar regions occurs at both synoptic and planetary scales.

### 4.1.3  Planetary transport

The planetary energy transport is similar in both hemispheres, different from quasi-stationary transport which is mainly relevant in the NH (Figs. 1a, 4, 5a). The latter is

in agreement with Trenberth and Stepaniak (2003a), underscoring that quasi-stationary transport is a primary factor in the extra-tropical NH. The latter study associates this quasi-stationary transport with the planetary scale, albeit without proof – a hypothesis confirmed by this study (Fig. 4). A new finding that could partly be inferred from Fig. S3 of Lembo et al. (2019) is the near-symmetric structure of the planetary energy transport in both hemispheres. This symmetry could not have been anticipated by considering only quasi-

stationary transport since the planetary transport in the SH is mainly transient (Fig. 4), in agreement with Mo (1986).

In both hemispheres, the planetary transport is similar in the sub-tropics and lower mid-latitudes. In the higher mid-latitudes and polar region of the SH, it is approximately 20 % weaker than in the corresponding NH regions. Hence, eddies at similar spatial scales are transporting the energy in both hemispheres (see also Fig. 2), likely due to similar physical mechanisms in both hemispheres forming the energy-transporting eddies.

Generally, two patterns of planetary transport are identified:

1. In the sub-tropics, most planetary transport is associated with quasi-stationary moisture transport, $\widetilde{vQ}_{\mathrm{p}}^{\mathrm{q\text{-}s}}$. These waves presumably represent sub-tropical high-pressure and monsoon systems, which can prevail for several weeks. The planetary moisture transport leads to humidity divergence in the tropics and convergences in the sub- and extra-tropics (Fig. 5e).

2. In the extra-tropics, planetary eddies mainly transport dry energy and only little moisture. In the polar regions, this planetary energy transport is almost as important as synoptic transport. The peak in energy transport by planetary waves, $\widetilde{vE}_{\mathrm{p}}$, is at the polar boundary at a latitude of around 60° in both hemispheres (Fig. 5a, c), hence further poleward than the peak in synoptic transport at 45°. Thus, the planetary dry transport, $\widetilde{vD}_{\mathrm{p}}$, leads to energy divergence in the sub-tropics and mid-latitudes and convergence in the polar regions (Fig. 5f). These planetary waves are mainly transient (Fig. 4: 70 %) in the mid-latitudinal SH, whereas the quasi-stationary component dominates (60 %) in the corresponding NH region, which agrees with Peixoto and Oort (1992). However, in the high latitudes of the SH, a considerable fraction of the planetary transport is quasi-stationary.

### 4.1.4 Meridional overturning circulation

The meridional overturning circulation, $\widetilde{vE}_{\mathrm{md}}$, and its role in transporting energy poleward have long been known (e.g. Hadley, 1735; Lorenz, 1967; Peixoto and Oort, 1992). The Hadley circulation in the tropics dominates the energy transport in that region. The total energy transport, $\widetilde{vE}_{\mathrm{md}}$, by the Hadley circulation is small compared to the thermally direct dry component, $\widetilde{vD}_{\mathrm{md}}$, and to the thermally indirect latent component, $\widetilde{vQ}_{\mathrm{md}}$, since the latter largely compensates the former (Fig. 5a–c). Further, the annual-mean transport by the Hadley circulation is small since transport into the tropical winter hemisphere originates from the equatorial summer hemisphere, leading to compensation of summer and winter cells (see Sect. 4.2).

In the mid-latitudes, the meridional circulation, $\widetilde{vE}_{\mathrm{md}}$, features the thermally indirect Ferrel cell with a peak around

53°. The Ferrel cell is almost twice as strong in the SH than NH. The transport by the Ferrel cell is mainly in the form of dry energy, which is to a small extent compensated by thermally direct moisture transport. In the NH, the eddy-driven Ferrel cell (Vallis, 2017), spans the whole extra-tropics, including the Arctic. In the SH, a thermally direct polar cell is evident in the dry transport, $\widetilde{vD}_{\mathrm{md}}$, which is primarily driven by katabatic flow from Antarctica as noted by Trenberth and Stepaniak (2003a). Different to previous studies by for example Peixoto and Oort (1992) a NH polar cell is not evident in the here-utilised ERA5 dataset, neither in the annual mean nor in the summer or winter season (Sect. 4.2). In the Arctic, energy is transported by synoptic and planetary eddies favouring the formation of a Ferrel cell, whereas zonal symmetric katabatic flows, as observed in the Antarctic, do not develop due to the lack of a large ice dome centred over the pole.

### 4.1.5 Mesoscale transport

In contrast to synoptic and planetary waves, mesoscale waves, $\widetilde{vE}_{\mathrm{ms}}$, at scales smaller than 2000 km, are only responsible for a negligible part of the energy and moisture transport at all latitudes (Figs. 2, S7), in accordance with Graversen and Burtu (2016), who show that wavenumbers 0–10 are responsible for the majority of the energy transport at all latitudes. Atmospheric models may have larger challenges in reproducing mesoscale eddies as compared to eddies at larger scales. However, ERA5 at a horizontal grid spacing equivalent to 30 km can reproduce mesoscale cyclones, such as polar lows (Stoll et al., 2021), and hence the negligible importance of mesoscale eddies for the total energy transport appears trustworthy. Due to its negligible role, we include the mesoscale into the synoptic transport for the remainder of this study such that the sum of all scale components yields the total transport.

## 4.2 Seasonal transport

Some transport patterns become more apparent when seasons are analysed separately. The NH summer and the SH winter are defined as June to August, and the NH winter and the SH summer are December to February. In spring and autumn the energy transport is mainly similar to the annual-mean transport (Figs. 5, S8).

Due to seasonal variations in the thermal contrast between the Equator and poles, more energy is transported poleward in the winter than in the summer hemisphere (Fig. 6a, d) in good agreement with previous studies (e.g. Peixoto and Oort, 1992). The seasonality in the transport is larger in the NH than the SH, as also noted by Trenberth and Stepaniak (2003a). This is affected by a larger annual temperature cycle in the NH due to its large continents having a smaller heat capacity than the oceans in the SH. Still, the two hemi-

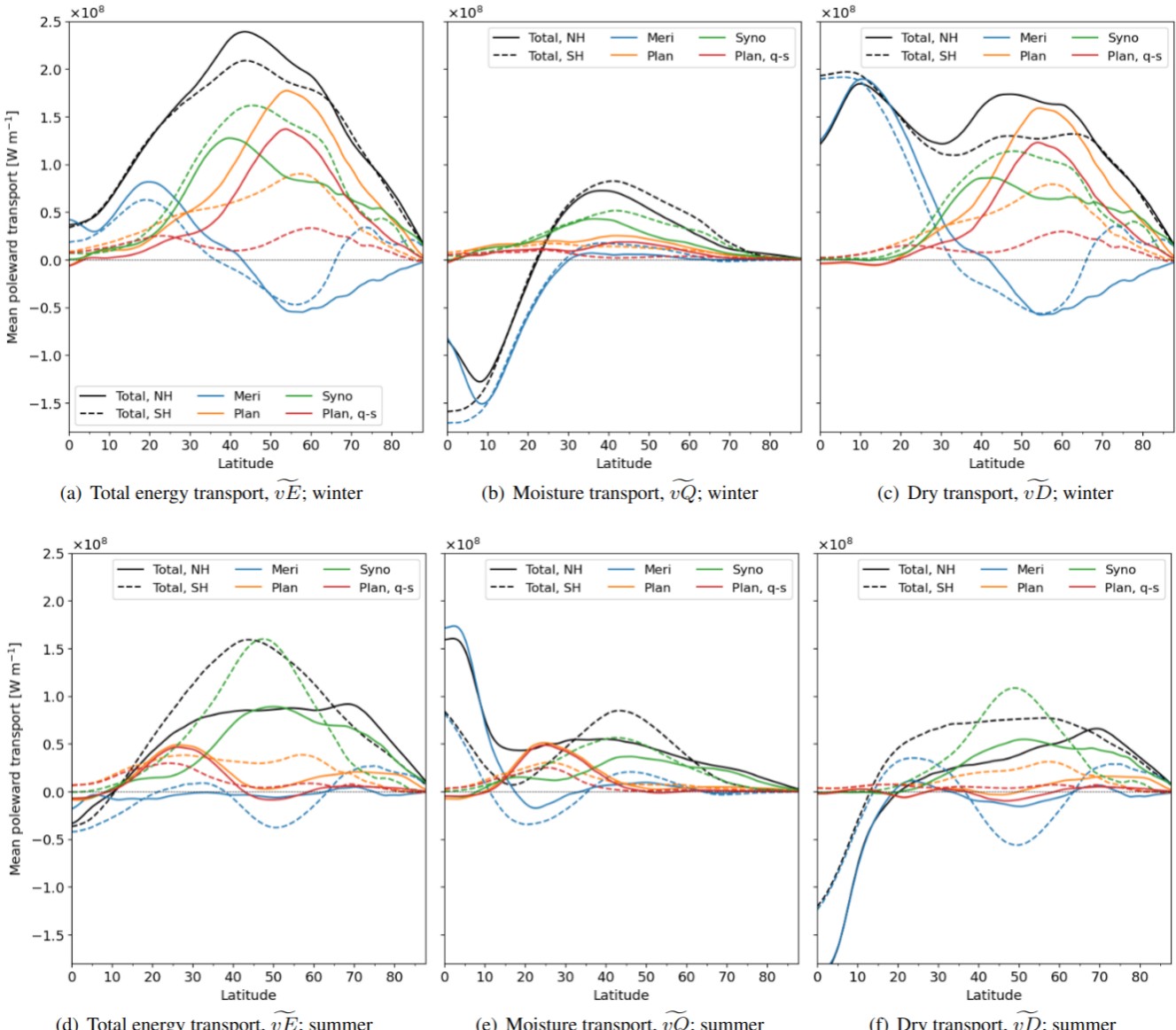

**Figure 6.** As Fig. 5a–c but for the seasonal-mean transport of **(a–c)** winter and **(d–f)** summer for **(a, d)** the total energy transport, **(b, e)** its latent component, and **(c, f)** its dry component.

spheres share many characteristics in the atmospheric energy transport regarding seasonality.

The location separating northward and southward total transport, the energy flux equator (Adam et al., 2016), is at around a latitude of 10° in the summer hemisphere (Fig. 6d). This is close to the ascending branch of the Hadley circulation (Vallis, 2017, p. 514). In the tropics, the meridional overturning circulation, $\widetilde{vE}_{\mathrm{md}}$, is most important (Fig. 6, blue). It transports energy from the summer tropics to the winter sub-tropics, whereas moisture, $\widetilde{vQ}_{\mathrm{md}}$, is transported in the opposite direction.

In the extra-tropics of both hemispheres, the synoptic transport, $\widetilde{vE}_{\mathrm{sy}}$, is little influenced by seasonality, in broad agreement with results from wavenumber separated transport of the NH mid-latitudes in Lembo et al. (2019) and with transient transport in Trenberth and Stepaniak (2003a). In contrast, the planetary transport, $\widetilde{vE}_{\mathrm{p}}$, has a strong seasonality, being highly relevant in winter but almost absent in summer. Hence, the summer is dominated by synoptic transport, $\widetilde{vE}_{\mathrm{sy}}$. Differently, in winter, both planetary and synoptic waves are highly relevant for the energy transport. In the NH winter, planetary waves, $\widetilde{vE}_{\mathrm{p}}$, contribute to more transport than do synoptic waves, mainly by their quasi-stationary component, $\widetilde{vE}_{\mathrm{p}}^{\mathrm{q\text{-}s}}$. In the SH winter, planetary transport is also important but mostly in its transient form, $\widetilde{vE}_{\mathrm{p}}^{\mathrm{tr}}$, whereas its stationary part is small as noted previously (Oort and Peixóto, 1983).

In the sub-tropics of the summer hemisphere, quasi-stationary planetary waves, $\widetilde{vE}_{\mathrm{p}}^{\mathrm{q\text{-}s}}$, are among the largest contributors to poleward energy transport (Fig. 6d). These waves are transporting energy mainly in the form of moisture, $\widetilde{vQ}_{\mathrm{p}}^{\mathrm{q\text{-}s}}$ (Fig. 6e), presumably reflecting the summer monsoon and long-lasting sub-tropical high-pressure systems.

## 4.3 Inter-annual variability

In this study, the inter-annual variability in the energy transport is computed by the standard deviation of the annual-mean transport. The total energy transport, $\widetilde{vE}$, is only varying by a few percent between years for all latitudes (Fig. 7a). However, the inter-annual variability in the individual components is up to 3 times larger than the variability in the total transport, especially in the extra-tropics (Fig. 7a). Hence, large transport in one component is typically compensated by smaller transport in another, as noted by Trenberth and Stepaniak (2003a) and Lembo et al. (2019).

In contrast to the total energy transport, the variability of the total moisture transport, $\widetilde{vQ}$, is larger than the variability of its individual scale components (Fig. 7b). Hence, the moisture transport components are not compensating each other in the same manner as the total energy transport components. Instead, the compensation of the components of the total energy transport is in the form of dry energy (Fig. 7c).

We hypothesise that the different co-variability of the scale contributions for the total energy and moisture transport is due to their different underlying mechanisms. Preliminary results point towards the annual-mean energy transport being induced by the meridional temperature gradient in the manner of a diffusion process with a globally almost constant diffusion coefficient. Hence, large transport in one component reduces the temperature gradient, leading to less transport in another. Differently, moisture is a tracer of the atmospheric circulation and, therefore, not described by a diffusion process such that the components do not compensate similarly to the total energy transport.

The tropics feature large inter-annual variability in the moisture transport, $\widetilde{vQ}$, and approximately equally large variability in the dry transport, $\widetilde{vD}$ (Fig. 7b, c). The large total variability in the moist and dry energy transports is almost entirely due to variability in the meridional overturning. This is not surprising since the meridional overturning is responsible for most of the moist and dry energy transport in the tropics (Figs. 5b, c, 6b, c, e, f). However, the variability in total transport, $\widetilde{vE}$, is only a fifth of the moisture transport variability (Fig. 7a, b). CE6 Hence, a Hadley circulation causing a larger-than-usual thermally direct dry energy transport in a single year also causes a larger-than-usual thermally indirect latent energy transport in that year. This explains that the total energy transported by the Hadley circulation is more or less constant between years.

In the extra-tropics, the planetary transport, $\widetilde{vE}_{\mathrm{p}}$, is the component exhibiting the largest inter-annual variability (Fig. 7a). This is remarkable since the annual-mean planetary transport is mainly smaller (Fig. 5) than the synoptic transport, $\widetilde{vE}_{\mathrm{sy}}$. Consequently, the planetary transport, $\widetilde{vE}_{\mathrm{p}}$, varies from year to year by approximately 10 % in the mid-latitudes and 15 %–20 % in the polar regions. In contrast, the synoptic transport, $\widetilde{vE}_{\mathrm{sy}}$, varies only by around 5 %. The large planetary variability is mainly attributed to its quasi-stationary component, $\widetilde{vE}_{\mathrm{p}}^{\mathrm{q\text{-}s}}$, whereas its transient component, $\widetilde{vE}_{\mathrm{p}}^{\mathrm{tr}}$, is much less variable (not shown). This demonstrates the "quasiness" of the quasi-stationary transport, being the transport component that varies most from year to year. This presumably reflects the role of (inter-)annual modes of climate variability in favouring different quasi-stationary circulations and planetary transports in different years.

The extra-tropical moisture transport, $\widetilde{vQ}$, is varying approximately equally for planetary, $\widetilde{vQ}_{\mathrm{p}}$, and synoptic waves, $\widetilde{vQ}_{\mathrm{sy}}$ (Fig. 7b). Since synoptic waves are responsible for most of the moisture transport in the extra-tropics (Fig. 5b), the variability fraction is considerably higher for the planetary than the synoptic moisture transport.

## 5 Discussion and conclusion

In this study, we analyse the global atmospheric circulation by separating the meridional energy transport of the years 1979–2021 in the ERA5 reanalysis by spatial scales, by moist and dry components, and by quasi-stationary and transient parts. For separating the energy transport by scales for all latitudes, we apply a new approach by using the wavelength instead of the wavenumber utilised previously (e.g Graversen and Burtu, 2016).

We demonstrate that separating transport by synoptic and planetary eddies at a wavelength of 8000 km reflects the physically grounded distinction between baroclinically induced and other eddies. Moreover, we show that the wavelength of 8000 km is reasonable for separating at all latitudes. This separation wavelength approximately agrees with the conventional separation between transient and quasi-stationary eddies, as most eddy transport smaller than 8000 km is transient, whereas most quasi-stationary transport occurs at the planetary scale larger than 8000 km. Despite the latter, a considerable amount of the planetary energy transport is transient, especially in the extra-tropical SH.

It should be noted that the scale separation is implemented continuously so that the largest eddy with a wavelength smaller than 8000 km is partitioned between the planetary and synoptic scales. This implies that the planetary transport includes waves of around 8000 km in size, whereas synoptic transport is by waves strictly smaller than 8000 km.

Eddies with a wavelength of almost 8000 km may appear as large for being part of the synoptic scale. However, most baroclinically induced and most transient energy transport organises at wavelengths around 5000 km at all lati-

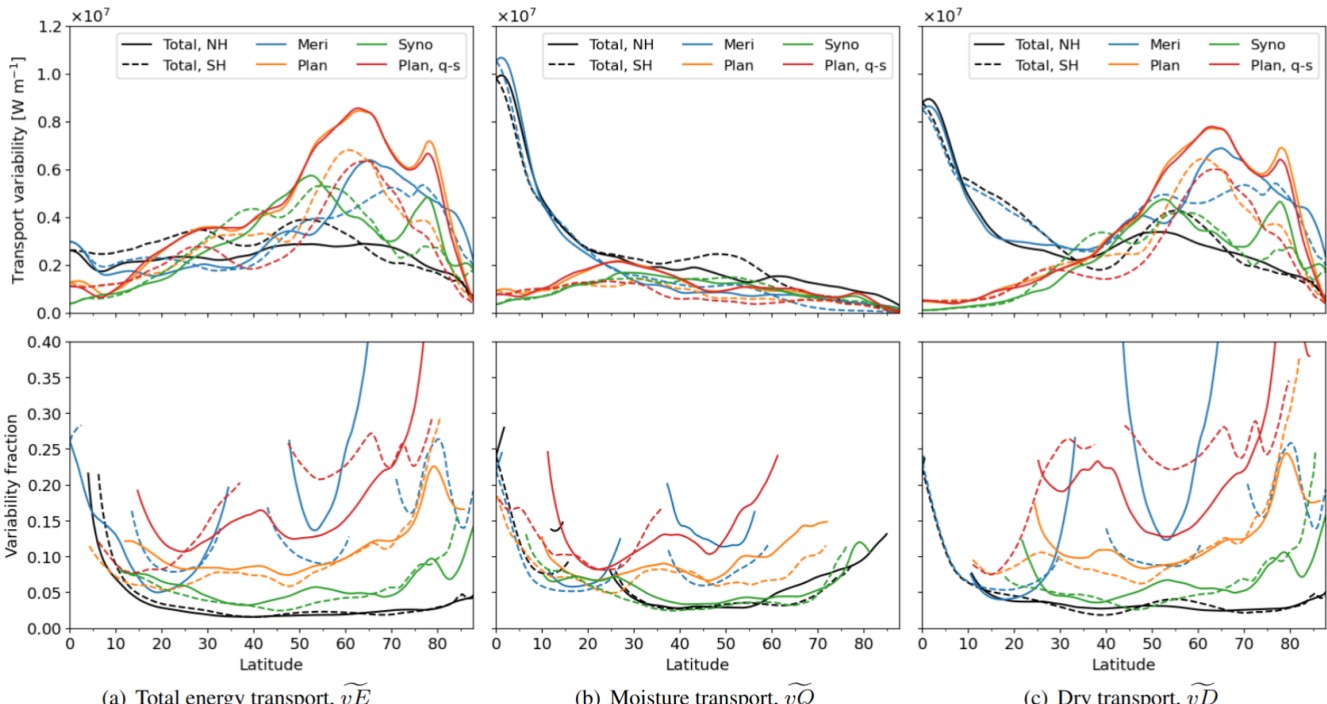

**Figure 7.** (**a**, upper panel) Inter-annual variability in the meridional energy transport, computed as the standard deviation in the annual transport, from ERA5 for the years 1979–2021 (black). The energy transport is divided into the meridional overturning circulation (blue) and wave components at the planetary (orange) and synoptic (green) scales. Solid and dashed lines depict the Northern Hemisphere and Southern Hemisphere, respectively. (**a**, lower panel) The fraction of the variance of each transport component related to its absolute mean transport. Values are masked at latitudes ±5° where the mean transport crosses zero or where the absolute mean is smaller than 5 % of the zonal maximum of the total transport. (**b, c**) As panel (**a**) but for the latent and dry energy transport.

tudes (Figs. 2, 3), agreeing well with the predicted length by dry-baroclinic theory (Vallis, 2017). Despite the maximum at around 5000 km, the baroclinically induced and transient energy transport occurs in a wavelength band between approximately 2000 and 8000 km; hence separating at around 5000 km would be misleading.

Synoptic eddies are perceived as low- and high-pressure systems near the surface CE7 and small-scale Rossby waves that are often strongest around the tropopause. In the baroclinic development, near-surface pressure systems vertically interact with the upper-level Rossby waves. One synoptic wave includes both a low- and a high-pressure system. Hence, a 2000–8000 km range implies that synoptic cyclones and anticyclones feature a typical diameter of between 1000 and smaller than 4000 km and that the typical distance between the cores of two independent (anti)cyclones is between 2000 and 8000 km. Both appear reasonable from comparison with weather maps. Even in situations of cyclone clusters, arguably situations where the distance between cyclones is unusually low, the spacing between cyclonic centres is around 2000 km (e.g. Weijenborg and Spengler, 2020, Fig. 2d–f), the lower end of the here-defined synoptic scale. Note that the cyclones of that cluster share a common front and hence are not independent. The distance of the cyclones is in the north-

eastward direction, whereas the closest cyclones in the zonal direction are at larger distances of around 5000 km. Hence, our methodology relying on a zonal Fourier decomposition appears accurate for such situations.

A single, sharp spatial separation between the synoptic and planetary scales may not exist in the real world. However, a spatial decomposition is helpful for better understanding the atmospheric circulation and its impact on regional climate (Baggett and Lee, 2015; Graversen and Burtu, 2016; Röthlisberger et al., 2019). Here, we provide further arguments for the usefulness of separating the energy transport by spatial scale.

Different from the classical separation into quasi-stationary and transient energy transport, the spatial scales of the transport are quite similar in both hemispheres, pointing towards similarities in the contribution of active physical mechanisms. The most pronounced difference between the two separation methods is that planetary transport, $\widetilde{vE}_{p}$, is broadly similar in both hemispheres, whereas quasi-stationary transport, $\widetilde{vE}^{q\text{-}s}$, is mainly important in the NH.

In the annual mean, most energy and moisture in the extratropics are transported by synoptic eddies, $\widetilde{vE}_{sy}$. It is remarkable that in the large range of atmospheric eddies, those at scales in the band between 2000 and 8000 km are responsi-

ble for most of the meridional energy transport for the whole extra-tropics. This points towards the importance of baroclinic instability for inducing the extra-tropical energy transport as long hypothesised (Holton and Hakim, 2013). However, other mechanisms may contribute to the formation of eddies at the synoptic scale. Hence, a future study is planned to investigate the causes and effects of energy transport at different scales.

The synoptic energy transport reveals to be influenced little by the season. In contrast, extra-tropical planetary transport, $\widetilde{vE}_\mathrm{p}$, is of major importance only in the winter season mainly by transporting dry energy, $\widetilde{vD}_\mathrm{p}$. In winter around the Arctic boundary, quasi-stationary planetary waves, $\widetilde{vE}_\mathrm{p}^{\mathrm{q-s}}$, dominate the energy transport. Such quasi-stationary planetary waves around the polar boundaries of both hemispheres feature the transport component with the largest inter-annual variability globally.

Other known characteristics of the atmospheric circulation are reproduced in this study, such as the dominance of the Hadley circulation in the tropics for transporting energy, $\widetilde{vE}_\mathrm{md}$, to the winter hemisphere and transporting moisture, $\widetilde{vQ}_\mathrm{md}$, in the opposite direction. In the sub-tropical summer, quasi-stationary planetary waves transporting moisture, $\widetilde{vQ}_\mathrm{p}^{\mathrm{q-s}}$, are the largest contributor to poleward energy transport, which is associated with monsoon systems.

In this study, the atmospheric transport is analysed on an annual-mean and seasonal-mean basis, whereas on shorter timescales, the transport can be highly sporadic and displays large deviations from climatology (Swanson and Pierrehumbert, 1997; Messori and Czaja, 2013; Lembo et al., 2019). Some planetary transport events in the extra-tropics are for example equatorward. Hence, when the seasonal-mean poleward planetary transport is close to zero, this can mean either that the planetary transport is generally small or that poleward and equatorward transport events are balancing each other. A follow-up study will investigate the intra-seasonal distribution of the different transport components. Another option for investigating the spatio-temporal scale of eddies is the usage of Hayashi spectra as in Dell'Aquila et al. (2005) that performs a Fourier decomposition in both space and time.

*Data availability.* The computed decomposition of the energy transport based on ERA5 and the code for the analysis are willingly provided on request.

*Supplement.* The supplement related to this article is available online at: https://doi.org/10.5194/wcd-4-1-2022-supplement. TS3

*Author contributions.* RGG calculated the Fourier-decomposed energy transport. PJS performed further data processing and visualisation. PJS TS4, RGG, CE8 and GM contributed to interpreting the results and writing the manuscript.

*Competing interests.* The contact author has declared that none of the authors has any competing interests.

*Acknowledgements.* Thanks go to ECMWF for providing access to data from the ERA5 reanalysis. The data were partly processed with the supercomputer Fram and stored at NIRD, both provided by the Norwegian Research Infrastructure Services (NRIS) Sigma2 AS (project nos. NN9348K and NS9063K, respectively). We also thank two anonymous reviewers and the editors for their constructive comments, which considerably improved the paper.

*Financial support.* This research has been supported by the Research Council of Norway (NFR) under the projects "The role of the atmospheric energy transport in recent Arctic climate change" (no. 280727) and "Stability of the Arctic climate" (no. 314570).

*Review statement.* This paper was edited by Camille Li and reviewed by two anonymous referees.

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

**Remarks from the language copy-editor**

CE1 Please note the use of "waves" for consistency within the paper.

CE2 Please note that "is" was retained for consistency within the sentence.

CE3 Please confirm that "firstly", "secondly", and "thirdly" have been properly inserted.

CE4 Please confirm that "firstly", "secondly", and "thirdly" have been properly inserted.

CE5 Please note the change to "a" for consistency within the paper.

CE6 Please note the insertion with minor edits.

CE7 Please note the removal of the comma here in line with our standards.

CE8 Please note the insertion of a serial comma here in line with the rest of the paper.

**Remarks from the typesetter**

TS1 Please check all affiliations carefully.

TS2 Please note that "Section(s)" is only written out at the beginning of a sentence as per our standards.

TS3 Please note that the Supplement cannot simply be replaced as you added a completely new section. Please provide the original Supplement file with the adjusted section numbering or provide a detailed explanation for those changes that can be forwarded to the editor. In a post-review process, the editor can check the changes and decide if the new supplement (with the additional section and figure) can be used. Thank you for your understanding.

TS4 Please confirm.

TS5 Please confirm.

TS6 Please check all names carefully as they were not provided in the correct format.

TS7 Please check.