# Peer review of "The global atmospheric energy transport analysed by a wavelength-based scale separation"

_Weather and Climate Dynamics, 2022_

## Referee Comment (RC2)

**General comments**

Overall the authors present a nice study about the global atmospheric energy transport, based on different scale separations. The authors argue that a wavelength-based spatial scale separation, compared to the wavenumber-based separation, is useful to better understand the atmospheric circulation and its impact on the local climate. The authors focus on an annual mean analysis, but also checked how their results vary interannually and for different seasons. The results, separated by different scales for both the northern and southern hemisphere, does include the total energy transport, as well as the moisture and dry static transport part.

In general, I think the presented arguments and discussion in this manuscript could be a bit clearer. A stronger focus could also be on the dynamical understanding of the contribution of the different spatial or temporal scales. I think this is important, as the goal of the paper is to use the somewhat new wavelength based consideration to get a better understanding of the atmospheric circulation. Maybe the authors could also make it clearer what exactly is the new contribution of their study , because this wavelength vs wavenumber consideration was also discussed in previous studies. It is not fully clear to me, if it is the application to the energy transport analysis, the more in detail and systematic approach of the comparison with more traditional scale separation or if the authors see this procedure as something fully new. In terms of getting a deeper understanding, I would also suggest to give more context and insight into the sensitivity test, e.g. the sensitivity on the choice of the wavelength scale is only done for the climatological annual mean signal as well as for the impact of the quasi-stationary contribution to the synoptic scale, but the results are then analysed for different seasons without discussing how those time and spatial scale choices might impact those results. In particular the neglect of the quasi-stationary synoptic signal should be discussed in more detail, as especially for different seasons one might expect a shift of the contributions from the planetary to the synoptic scale or the quasi-stationary to the transient scale, dependent on how the threshold was chosen. I have the feeling the reader might not get a much deeper understanding from the authors introduced wavelength based consideration, as those mentioned aspects are not really explained sufficiently and the reduced sensitivity testing might leave the reader with several open questions. E.g. in the conclusions the choice of the wavelength threshold is said to be based mainly on the intuition of the authors, which is not very convincing. Without a clearer sensitivity testing (which should also be discussed in the conclusions) it might be hard to convince the reader of this approach. So in general, I think, highlighting the different approaches is already very useful, so therefore this manuscript is already quite useful. But I think the authors should slightly improve in the presentation of their results and put more effort into highlighting what deeper understanding the reader can get from this approach. This also includes a more convincing sensitivity study to better understand the impact of the threshold choices on the differences between the scale contributions, not only for the annual mean signal but also the different seasons , because that is when I would expect the largest impact (e.g. dominant wavenumbers in midlatitudes are quite different in

summer and winter). In the following I included further specific comments.

**Specific comments**

**Introduction**

**line 30:**
"fast-varying" means everything faster than a month here? As the authors highlight different disturbances, such as polar lows, maybe faster varying would be a better description as also much slower disturbances are part of the same "fast-varying" group.

**lines 38-41:**
This sentence somehow seems to suggest that the previous studies were missing an important point, i.e. the latitude dependent spatial scale of a wavenumber. However, this might not be that relevant for their studies, e.g. Röthlisberger et al. (2019) are interested in the occurrence of wave patterns in the midlatitudes. As they are not primarily interested in the meridional transport of energy and how the wave separation changes with latitude, this seems to be another question and is therefore not necessarily a disadvantage of their method. I would suggest that the authors make this point clearer, i.e. using a fixed wavenumber range might be fine for studies that look at specifics dynamics at a fixed latitude range, but it could be misleading if one does investigate the dynamics across a large range of latitudes.

It might be worth mentioning that similar studies of wave patterns in the midlatitudes, based on wavenumbers, did account for this by varying the wavenumber depending on the latitude, e.g. Wolf and Wirth (2017, Diagnosing the horizontal propagation of Rossby wave packets along the midlatitude waveguide, see their Fig. 6). So using a method based on wavenumbers, does not necessarily mean that one cannot account for this effect.

**line 54:**
Referring to: "However, transport at other scales could be of transient character as well." and "In this study, we are pointing out that the separation..." I would suggest the authors be more specific here and more clearly about the point they try to make or slightly reformulate this paragraph. This paragraph seems to suggest that this point (other scales can be of transient character, too) is an important new point tackled in this study. However, there are studies which explicitly highlight the point that quasi-stationary waves can be transient, as it is already part of their name (quasi-stationary, so not necessarily stationary). So I think the authors should highlight their real contribution better, namely doing this more systematically. Although many other studies also highlight this point, or modify their wavenumber based method to account for the latitude dependent spatial scale effect, this study is systematically investigating this issue in more detail.

**line 61:**
I would suggest to mention all following sections here, not only the data section.

**Data and methods**

**line 71:**
Maybe "In this study, we take a zonal-mean perspective of the local atmospheric energy transport, ..." to make it right away clearer to the reader that this is a local approach.

**lines 71-74:**
Maybe the authors can be more specific here about the differences of the two approaches, as from this description I don't really see the difference between a zonal mean or the zonal integral. Maybe I am missing an important point, but not fully obvious to me why I should expect the peaks at different latitudes.

**line 76:**
ylabel: length instead of lenth

**line 81:**
Why is this extensive smoothing necessary, so why not only before the calculation of derivatives (to get rid of possible large unrealistic gradients) but also afterwards again?

**lines 86-87:**
Is this formulation ("only possible from a time-mean perspective") justified? In general the quasi-stationary transport does not need to be based on monthly mean fields, so if this would be adapted then it would also be possible on smaller temporal scales. Further, the authors mention that other studies are "normally" basing this on monthly fields, which suggests that not all are following this procedure, which would not support the "only possible from" formulation.

**line 91:**
Only $v$ appears in this equation, not $\mathbf{v}$, so I would suggest not to refer in the notation explanation to the full wind.

**line 93:**
Maybe the authors could be more explicit by what they mean when they say "in three ways that can be applied in combination". Those are three different ways to identify energy transport, so what do the authors have in mind if they talk about combining those separations?

**line 94:**
Replace "archived" by "achieved".

**line 104:**

Replace reference to "Fig. 1a" by "Fig. 1".

**lines 117-118:**
Equation (7) represents the zonal mean for one particular time step, correct? Figure 2 however shows a temporal average, so I would suggest to explicitly mention this in lines 117-118 so that the reader does not get confused with the notation ($[\cdot]$ only representing zonal mean).

**line 126:**
Continuous separation shown in Fig. S2b, not S2a.

**Wavelengths utilised for scale separation**

**lines 144-145:**
I find it difficult to follow the conclusion for the use of a band between 2000-8000 km. The reference threshold is about 4000-4700 km, which would correspond to a wavenumber of 6-7 at about 45° latitude. According to the authors there is some variability associated with the scale of synoptic cyclones and in the fourier decomposition also neighbouring wavenumbers contribute strongly to the energy transport. But 2000 km (8000 km) at 45°N would be represented by a wavelength between 14 and 15 (3 and 4) and those larger scales could also be associated already with stationary or quasi-stationary longitudinally extended waves. Those wave patterns can still be associated with the smaller than planetary wave scales and therefore be considered as part of the smaller synoptic band, but the authors explicitly highlight the link to the much smaller synoptic cyclones. To me it is not clear from this paragraph what the authors want to have included in their synoptic band. Also from the sentence with the reference to the synoptic Rossby waves (lines 145-146) it is not clear if the authors want to have those included or if they just tolerate this to be able to capture most of the energy transport with the synoptic scales they are interested in. I suggest the authors to rephrase this paragraph to make this clearer. If the authors are indeed only interested in the smaller scales as synoptic cyclones, I think there is some more justification necessary for the choice of the upper threshold of 8000 km, because I would expect no synoptic cyclone at 45°N would be represented by a wavenumber 3 to 4.

**lines 198-199:**
Why is this contribution (quasi-stationary component of synoptic transport) not further investigated? The reason seems to be that it its contribution to the synoptic scale is rather small (although up to 30 %) and doesn't really fit into the category of quasi-stationary planetary scale? Excluding this contribution seem to suggest that the introduced categories of synoptic and planetary or transient and quasi-stationary have some difficulties capturing the processes they are supposed to capture. This part therefore could also be seen as some measure of categroy uncertainty, excluding it fully seems a bit surprising.

**lines 204-205:**
Maybe the authors should specify here a bit more to which main results they refer, as the lines for different wavelengths are very different, e.g. in terms of the contribution from synoptic and planetary scales in midlatitudes (synoptic much stronger for 10000 km, but weaker for 6000km). So the wavelength has a huge impact on the separation between planetary and synoptic scales (qualitative different conclusions). Therefore I think the authors should give some more context here, for what results/analysis this separation does not matter.

**Organisation of the global energy transport**

**lines 211-213:**
I would suggest to avoid the reference to Fig. S6, as this figure is about the impact of using different scales. Further Fig. 4 is the relevant figure, which shows the signal for both hemispheres, so there does not seem to be any need to additionally refer to another figure. Further, is a particular reason that only the x-axis for panel d to f is scaled, but not for panel a to c? I found it initially a bit confusing when I tried to compare transport and convergence fields.

**line 216:**
I would exclude "almost", because the curves are similar in the sense that they have the shape. I guess almost refers to the amplitude difference, but this is explained in the following.

**lines 225-226:**
Do the authors really mean an inverse sine function in Fig. 4b? I find it hard to identify this curve behaviour in this plot. Further isn't there a difference between NH and SH (next sentence in lines 226-227 seems to suggest this is not the case)?

**line 229:**
Maybe refering to the curves as showing a plateau is a bit too much, at least for both hemispheres. Maybe this can be rephrased slightly with saying "more plateau-like" or something similar.

**lines 249-250:**
But isn't that what Fig. 4a is showing, that plan q-s is much stronger in NH than SH? This statement refers to previous interpretation of this manuscript or other studies (Trenberth and Stepaniak, 2003)? I don't really understand the contradiction, because the statement seems to agree with the figure. If the contradiction refers to the similar curves of the planetary signal for both hemisphere, then I also don't understand the contradiction, because individual parts of this signal (q-s and transient) do not necessarily need to have the same behaviour. I would suggest to rephrase this paragraph to make this clearer.

**lines 260-262:**

Why is this exactly most remarkable, or as this agrees with other studies do the authors have an explanation? As the strength of the wave guides in the NH and SH are very different, with the SH having a stronger jet, a separation by an identical time filter for identical wavenumber signals would lead on one hemisphere (SH) to the identification of a transient signal whereas on the other hemisphere (NH) as a quasi-stationary signal.

**lines 295-296:**

This is linked to a previous point, refering to lines 198-199. During winter there is a stronger wave guide and if considered in a power spectra spaned by wavenumber and latitude, more power at all latitudes is shifted towards smaller wavenumbers (compared to summer). This means, during summer the center of the power distribution will be located higher wavenumbers. It is therefore possible that the contribution of quasi-stationary signal is included in the synoptic scale. The authors mentioned in lines 198-199 that this part (quasi-stationary signal within synoptic scale) will not be considered as it does not represent such a large contribution. However, in summer this contribution could be larger. As the authors investigate all season (annual mean) and all seasons individually, I think they should be more specific about this point, e.g. when discussing this contribution in lines 198-199 they should already consider the seasonal differences. If this contribution would be larger in summer, their argument of not considering this contribution because of their small contribution seems more problematic. Further, the authors discuss the seasonal differences with Fig. 5 while excluding this part completely (synoptic and quasi-stationary). I would find it very interesting to see this contribution also included in those plots as it also shows the sensitivity of the analysis to the defined classes (synoptic, planetary, q-s, etc) and differences in the dynamics for the different seasons.

**lines 296-297:**

As mentioned in the previous point, it could also be that synoptic signals dominates in summer because the q-s is no longer mainly part of the planetary signal, but during summer part of the synoptic signal (but not considered). This somehow is strongly linked to the important point of this paper, highlighting the point that defining patterns on wavenumbers can be problematic because of the latitude dependence. The authors show convincingly the relevance of this point in great detail. But isn't it also relevant to consider the timescale of the wave patterns as function of the season, as it was for the spatial scale as function of latitude?

**lines 309-310:**

What do the authors mean by "are rather summing up to the total variance"? The signal is not summing up, but that is also not really expected that the variances sum up, as already stated by the authors in the previous sentence. The variability fraction also shows that the individual parts show stronger variability than the total signal, so there is the same kind of compensation between the different signals with overall smaller values. I would suggest the authors specify more in detail why this panel is so much

different to the other one to better support their statement and following hypothesis.

**lines 313-315:**
This statement is explicitly about the Meri-part? If so, the authors should make this clear, because the statement in its general form does not seem to be supported by Fig. 6.

**line 318:**
Is approximately 10 % correct? It seems that all values in the extratropics exceed 10 %, with values up to about 20 %.

**lines 320-323:**
This is again linked to my comments about lines 198-199 and 295-296. The planetary variability is strongly linked to the variability of the q-s component. If the q-s signal is linked to the strength and/or location of the wave guide, isn't it possible that part of it fall into the synoptic part for specific years? This would then be visible in the synoptic q-s part, but this is not part of this analysis here.

**Discussion and conclusion**

**line 332:**
I think it is not a really strong and convincing argument to base the choice of length scales on the intuitive understanding. For example, with my intuitive understanding I would have chosen a slightly different range of length scales. I understand that any choice will always be subjective, because there is no truth for doing the separation, but it can be stated like this or also refered to similar length scales in other studies, I would however suggest to not base the argument on intuition.

**lines 333-336:**
This spatial separation is such an important feature of the presented analysis. Therefore, I think the authors should include a comment here in the conclusion about the sensitivity. I included a statement in the result section about this sensitivty as well, which might be relevant here as well. I would include this sensitivity test (Fig. S6) even in the main manuscript and discuss the identified differences. If there are no relevant differences then I would agree to keep it in the supplementary material and just say that the results are not sensitive to the exact choice of length scale. However, as included in my previous comment about this issue, I think there are relevant differences. If the authors agree on this point, I think it makes sense to include it here in the main manuscript, if the authors disagree, then I think they should make it clearer what relevant part of the results are similar for the different length scales.

**line 342:**
Referring to "rather narrow band". Is this really a narrow band? This range represents a wavenumber range of about 3.5 to 14.2 at 45° latitude, which doesn't seem very narrow.

**line 347:**

I think it should be stated somewhere in the conclusion that the q-s part of the synoptic scales is excluded. I would further suggest to include a whole paragraph to discuss this exclusion, why it was done and what possible impacts could be for the results or the sensitivity of the study. How relevant is this excluded part for the different seasons?

---

## Author Response (AR1)

**Response to reviewer 1.**

We thank the referee for the thorough and constructive review. The comments are contributing to an overall improved manuscript. Note that some formulations in the new manuscript are slightly different than here since they were changed again in a proofreading process.

General response: *The manuscript "The global atmospheric energy transport analysed by a wavelength-based scale separation", by P.J. Stoll and R.G. Graversen (ID: wcd-2022-26) describes a wavelength decomposition of meridional energy transports in the atmosphere. Revisiting a common approach that has been often used in recent literature, overcoming the partitioning of eddies in a transient and quasi-stationary component and instead discerning between planetary, synoptic and mesoscale eddies according to their zonal wavenumbers, the authors emphasize the importance of distinguishing different thresholds of spatial scale separation for the different eddies as a function of latitude. The authors apply the proposed wavelength decomposition to the overall energy and its components, focusing on moisture and latent energy, discussing their annual mean features, the seasonal cycle and interannual variability. The manuscript focuses on the advantages of adopting this methodology, compared to previous ones, emphasizing the emergence of some crucial features of the dynamics, e.g. the role of planetary scale transports in the Southern Hemisphere.*
*Overall, I think that the manuscript is reasonably well written, contains an in-depth discussion of the caveats often overlooked when using a well-established methodology, and addresses some theoretical aspects of the general circulation of the atmosphere that, although not unprecedently seen, are enlightened in a clear and unambiguous way, allowing for potential development on these specific topics.*

Response: We thank for the overall positive feedback.

Reviewer: *What I find surprising, though, is that the authors do not actually focus on conveying in a convincing way neither the potential of the novel methodology, nor the implications for our understanding of the dynamics.*

Response: In the revised manuscript, we convey more clearly that the synoptic scale transport is largely associated to baroclinic eddies. Also in other aspects, we rewrite the manuscript in order to be more clear on advantages of the novel methodology.

Reviewer: *I was wondering if this may be due to a partial lack of context, and mistaking established facts as new findings. For instance, it is well known that meridional energy transport in midlatitudinal eddies is carried out by baroclinic instability mainly.*

Response: If the reviewer has references showing that baroclinc instability is carrying out the meridional energy transport we are interested to include them. We agree that baroclinic instability is responsible to form synoptic scale eddies in a manner to transport energy. However, also planetary eddies transport energy in the midlatitudes and these appear not to be of direct baroclinic origin (see new Figure 3).
Hence, we are not aware of a quantification of how much of the meridional energy
transport in the mid-latitudes can be associated directly with baroclinic instability.
Our separation now provides an estimate of such an quantification.

Reviewer: *Also, there are several works attempting to overcome the overlapping
notions of quasi-stationary waves and Rossby waves, by looking into Rossby wave
packets and local wave activity (Chang 2005; Grazzini and Vitart, 2014; Ghinassi et
al. 2018). Expanding on some hypotheses and considering available literature might
help overcoming the feeling of "speculative thinking" that sometimes underlies argu-
ments contained in the discussion (e.g. the statements about the role of monsoons
in summer planetary waves through moisture advection).*
Response: First of all, it would be helpful if the reviewer could specifically refer
to the literature in order to prevent misunderstandings. Three papers that we
associate with the mentioned ones, however do not appear to discuss "the overlap-
ping notions of quasi-stationary waves and Rossby waves". Surely, these interesting
works investigate wave activity, hence are related to the topic of our manuscript,
do, however, not focus on the main topic of our manuscript being the organisation
of the atmospheric energy transport. Hence, we would ask the reviewer to specify
in which respect our manuscript requires to refer to these works.

**Specific comments**

Reviewer: *ll. 32-43: This is one of the parts of the manuscript where I think that
the authors fail at defending the importance of the methodology they introduce. Two
aspects remain undiscussed: 1. The authors focus on zonal wavelengths, which is
perfectly understandable, but do not comment on what would happen if one would
consider meridional wavelengths, instead. 2. Their argument is in favor of choosing
scales partitioning wavelength-wise instead of wavenumber-wise, given the diversity
of scales across the latitudes. But there is nowhere shown that aspects of the trans-
ports that are emphasized with their methodology would not be seen when using a
"steady" wavenumber-based partitioning. A counter-factual example would help in
this sense;*

Response: To 1: We add considerable discussion along the meridional Fourier de-
composition:
"The Fourier decomposition is non-local, hence the whole circle influences the ob-
tained eddies. This makes the Fourier decomposition useful if the circle is governed
by similar eddies, which we observe from meteorological weather maps within the
different climate zones separated by latitudes. Theoretically a Fourier decomposi-
tion could be performed along longitude circles, e.g. along 0 and 180°. However,
a circle going around both poles and crossing the equator twice features eddies of
all climate zones, so it is questionable if we would gain useful understanding from
a meridional Fourier decomposition.
However, arguably the zonal scale is connected to the meridional scale of eddies,

becoming their general scale. From investigation of meteorological weather maps, we know (i) that synoptic-scale cyclones have an approximate similar zonal and meridional size since they are to first order circular, and (ii) that the meridional extend, i.e. the amplitude, of planetary Rossby waves, appears to roughly agree with the distance between a trough and a ridge, featuring half a zonal wavelength. Further, we show later (e.g. Fig. 2) that most of the mid-latitude transport occurs at zonal wavelength between 2000 and 8000 km, which is in broad agreement that events of extreme transport in the mid-latitudes are mainly coherent between 10 and 30° latitude (Lembo et al., 2019, Fig. 1d-g), considering that the event, such as a cyclone, has the size of half a zonal wavelength."

To 2: We emphasize two aspects that are misleading with a "steady" wavenumber-based partitioning: "Therefore the partitioning by wavenumber, for example between wave 3 and 4 as performed in many of the previously mentioned studies, leads to convergence of all eddy transport to the planetary scale towards the poles, whereas synoptic transport may be overestimated at low latitudes (Fig. 1b)."

Reviewer: *l. 126: same as above, the authors use the terms "wavelengths" and "spatial scales" almost in an interchangable way. I am a bit confused by this choice, as the claimed rationale behind this work is to capture the different scales of the eddy-driven transport at different latitudes.*

Response: Indeed, we use the wavelength to separate the spatial scales, so the terms are tightly connected. We state this more clearly by writing in line 39ff: "These studies separate the transport by a zonal wavenumber which can be associated with a zonal wavelength for a given latitude. As many of the previously mentioned studies, we interpret the zonal wavelength of the eddies as their spatial scale."

Reviewer: *ll. 180-183: I think this is one the main issues with the methodology here described. What latitude matters most for the definition of the eddy, the one where it starts to develop, the one where it grows, or where it decays. I think this has to do with the latitude at which the eddy is at its apex, and as a consequence transports more energy meridionally. The authors suggest here that the preferred spatial scale for synoptic scales relates to the latitude where the cyclogenesis occurs, i.e. the mid-latitudes. But then why do we need to care about latitude, in order to provide a relevant scale for separation between synoptic and planetary scales? This seems a bit of a contradiction, but it might be that I am missing something;*

Response: First of all this is just a hypothesis and we are not at all sure it is correct and do not have any prove. However, linear theory describing the growth of baroclinic eddies surprisingly well, is only valid in the initial baroclinic phase, as non-linear terms become large afterwards. Still, the linear theory well predicts the structure and scale of the evolving baroclinic eddies, hence we think that the initial phase is relevant for setting the size.

We do not understand where the reviewer sees a contradiction in our here formulated hypothesis that cyclones are propagating from the mid-latitudes to higher

latitudes in using the wavelength to identify the synoptic eddy transport at different latitudes. If the reviewer still sees a contradiction, we would be glad if (s)he could explain it.

However, we change the manuscript with the intention to make our argumentation more convincing. From: "It may appear surprising that the scale of maximum transient energy transport, $\widetilde{vE}^{tran}$, is independent of the latitude, since the deformation radius estimating the size of baroclinic eddies depends inversely on the Coriolis parameter, and depends linearly on the layer depth which decreases with latitude (Vallis, 2017). However, these are parameters important for the cyclogenesis which is mostly active in a confined region: Most cyclones originate from the mid-latitudes, where the horizontal temperature contrast is largest. The size of a cyclone is set during the genesis stage when the fastest-growing mode is prevailing. Many cyclones propagate to higher latitudes along the diagonal axis of the storm tracks (Shaw et al., 2016) and may keep their size."

Changed to: "It is surprising that the scale of maximum baroclinically-induced transport anomaly is independent of the latitude, since the Rossby deformation radius, $L_d = \frac{NH}{f}$, estimating the size of baroclinic eddies (Vallis, 2017), depends inversely on the Coriolis parameter, $f$, increasing with latitude, and depends linearly on the depth of the troposphere, $H$, and the tropospheric static stability, $N$, which mainly decrease with latitude. Hence, baroclinic eddies would be expected to be smaller at higher latitudes. A hypothesis for the latitude-independence of the baroclinic eddies is as follows: Most cyclones originate from the mid-latitudes, where the meridional temperature contrast is largest. The size of a cyclone is set during the genesis stage when the fastest-growing mode is prevailing. Many cyclones propagate to higher latitudes along the diagonal axis of the storm tracks (Shaw et al., 2016) and may keep their size."

Reviewer: *ll. 203-205: when comparing planetary scales and quasi-stationary components in Figure S6, it appears to me that the scale separation has to do with the scale of the maximum transient eddy activity (as shown in Figure 2), so that the larger the scale separation is, the more you find an overlap between quasi-stationary and planetary scales. As the separation scale gets smaller, the quasi-stationary component tends to vanish. This seems to me to suggest that quasi-stationary eddies are only those located in the ultra-long tip of the wavenumber spectrum, and the rest of the spectrum is mainly composed by transient waves. The two approaches to characterization of the eddies (wavenumber or time dependent) would then actually be coincident, for the right choice of the separation scale. Is that what you are aiming to show?*

Response: Indeed, planetary and quasi-stationary transport (as well as synoptic and transient) are partly overlapping, however, not similar as we demonstrate in Section 3.3.

Reviewer: *ll. 249-251: the relative symmetry of NH and SH planetary-scale transports is actually something new, to the best of my knowledge. I can think of some*

*similar results in the Supplementary Material of Lembo et al. 2019, but nowhere this was actually expanded. This is something that shall probably discussed, in terms of dynamical implications, in order to give a hint of how the methodology allows for a better understanding of the physical mechanisms;*

Response: Indeed, Lembo et al. (2019) is Fig. S3 compares the transport of both hemispheres and similarities can be recognised that are in agreement with our findings. We adapted the paragraph to include this and how it hints towards similar mechanisms in both hemispheres:

"The planetary energy transport is similar in both hemispheres, different from quasi-stationary transport which is mainly relevant in the NH (Fig. 1a, 5a). The latter is in agreement with Trenberth and Stepaniak (2003) pointing that quasi-stationary transport is a primary factor in the extratropical NH. They associate this quasi-stationary transport to the planetary scale, which they do not prove but which is confirmed by this study (Fig. 4). A new finding, that could partly be inferred from Fig. S3 of Lembo et al. (2019), is the almost symmetry of the planetary energy transport in both hemispheres, that could not been anticipated by the consideration of quasi-stationary transport since the planetary transport in the SH is mainly of transient character (Fig. 4). The planetary transport is similar in the subtropics and low mid-latitudes and only approximately 20% weaker in the higher mid-latitudes of the SH than the NH. Hence, eddies at similar spatial scales are transporting the energy in both hemispheres (see also Fig. 2), which is likely due to similar physical mechanisms leading to the energy transport."

Reviewer: *ll. 343-345: in the conclusion, the authors mention among relevant results that the extra-tropical meridional energy transport is mediated by baroclinic instability. But this is somehow known, and it has been shown, also analytically, in previous works. I can think, among others, of a few recent papers by Lenka Novak (Ambaum and Novak, 2014; Novak et al. 2015). As mentioned above, the authors evidence throughout the manuscript results that are genuinely new and potentially relevant, in order to understand the dynamics of heat exchanges (e.g. the role of planetary scales in the SH, of monsoons in moisture transport during the NH summer season). It is worth putting more emphasis on them in the conclusion as well;*

Response: Thanks for the positive perspective. Surely, it has been shown that baroclinic instability is responsible for a large amount of the eddy activity, however, we are not aware of studies that quantify the meridional energy transport of baroclinic eddies. We considerably rewrote the conclusions.

**Minor comments**

Reviewer: *l. 1: this sentence is more appropriate for an Introduction than an abstract. Consider removing;*

Response: As advised, we remove the sentence and replace it by "The atmosphere transports energy polewards in form of circulation cells and eddies."

Reviewer: *ll. 19-20: I am not entirely convinced that it should be stated in this way. The atmosphere is set in motion by rotation and angular momentum convergence as well, whereas it is clear that the atmospheric motions redistribute energy in order to contrast the differential diabatic heating between lower and higher latitudes;*

Response: We disagree that the atmosphere is set in motion by rotation and angular momentum convergence as well. These mechanisms clearly influence the motion of the atmosphere by, most importantly, the Coriolis force. However, the Coriolis force does not work at rest, so it can not set anything in motion. Would the atmosphere would be in rest, the largest term in the momentum equation is the pressure gradient force. The pressure gradient is set up by differential solar heating. Would our planet not receive differential heating, the first order terms in the momentum equation vanish, then lower order terms become relevant, such as the centrifugal force, and the motion in the atmosphere would arguably be quite different.

Reviewer: *l. 26: it is not entirely clear how the Hadley circulation appears in Figure 1, possibly some very quick description (as it is given below) could be provided;*

Response: We slightly changed the formulation: "In the tropics and sub-tropics, where the Coriolis effect is small, energy is predominantly transported by a zonally-symmetric meridional overturning circulation, known as Hadley cell (Hadley, 1735), and monsoon systems, organised by quasi-stationary cells (Fig. 1a)." The reader should be able to identify the meridional circulation in the figure since it is denoted in the legend.

Reviewer: *ll. 30-31: I wonder if the authors could expand on the definition of spatial scale here. In this work, it is often used as a synonym of "zonal wavelength", but the extent to which the interoperability of the two terms can be used is not clear to me;*

Response: At this point in the manuscript, we are still rather general and do not provide the "definition" of scale. However, in the following paragraph, we become more specific on our interpretation by adding the last sentence: "The scale separation of the meridional energy transport by a zonal Fourier decomposition became popular in recent years as it was applied to study the effect of energy transport for the Arctic (Graversen et al., 2021; Hofsteenge et al., 2022; Papritz and Dunn-Sigouin, 2020; Rydsaa et al., 2021), and for the mid-latitudes of the Northern Hemisphere (NH) (Lembo et al., 2019). These studies separate the transport by a zonal wavenumber which can be associated with a zonal wavelength for a given latitude. As many of the previously mentioned studies, we interpret the zonal wavelength of the eddies as their spatial scale."

Reviewer: *l. 61: a summary of the manuscript at the end of the manuscript is always needed, in my opinion;*

Response: A we added short outline of the manuscript: These questions are targeted in Sections 3 and 4. The main results are then summarised and discussed in Section 5. However, first the utilised data and methods are presented.

Reviewer: *ll. 65-66: the authors do not need to refer to ERA-Interim;*

Response: We removed that part of the sentence.

Reviewer: *ll. 70-71: not clear what the authors mean here, possibly rephrase;*

Response: We guess the reviewer mean ll. 71-72 and was not sure how we mean by the zonal-mean perspective. Hence we try to make the difference between the zonal integral and the zonal mean more clear by changing the formulation from: "In this study, we take a zonal-mean perspective of the atmospheric energy transport, which provides the transport through an atmospheric column with one metre width. Hereby, it provides a local measure of the transport, and differs from other studies that zonally integrate the transport along each longitude circle (Graversen and Burtu, 2016; Peixoto and Oort, 1992; Trenberth and Caron, 2001). However, the computed zonal integral of the energy transport from ERA5 (Fig. S1a) confirms the transport in these studies. For instance, the zonal-integrated poleward transport peaks at $4.8 \times 10^{15}$ W in the NH and $5.6 \times 10^{15}$ W in the SH at 41° latitude in both hemispheres. The latitude of maximum zonal-mean transport is slightly higher at 45°(Fig. S1b). Further, the average transport in the polar regions is more easily assessed by the zonal-mean transport as it is not influenced by converging latitudes."

To: "The zonal integral of the energy transport from ERA5 (Fig. S1a) confirms the transport in found in previous studies (Graversen and Burtu, 2016; Peixoto and Oort, 1992; Trenberth and Caron, 2001). For instance, the zonal-integrated poleward transport peaks at $4.8 \times 10^{15}$ W in the NH and $5.6 \times 10^{15}$ W in the SH at 41° latitude in both hemispheres. By computing the zonal integral of the energy transport, which depends on the length of the longitude circle, the transport becomes small at high latitudes since the longitudes converge (Fig. S1a). However, the local transport, expressed by the zonal mean, is considerable also in the polar regions (Fig. S1b). Hence, to compare the local importance of the atmospheric energy transport across all latitudes, we take a zonal-mean perspective which provides the transport through an atmospheric column with one metre width. Hereby, for example the latitude of maximum zonal-mean transport is at 45° latitude (Fig. S1b)."

Reviewer: *ll. 76-77: are the authors referring to geometrical constraints, when referring to "converging latitudes". If so? Please clarify why the zonal mean transport would be an advantage;*

Response: Indeed the longitudes converge. Thanks for spotting the mistake.

Reviewer: *l. 86: mentioning time-mean comparisons, it might be worth mentioning other decomposition techniques, allowing for space-time decomposition, e.g. 2-D wavelet decomposition or Hayashi spectra.*

Response: We add a sentence to the discussion that mentions the usage of the latter technique: "An option for investigating the spatio-temporal scale of eddies is the usage of Hayashi spectra as in Dell'Aquila et al. (2005) that performs a Fourier decomposition in both space and time." If the reviewer can point us to studies performing as 2D wavelet decomposition of the atmospheric dynamics, we would include them as well.

Reviewer: *l. 91: I have a few comments about the definition here. 1. why do you need to define the vector v if you are only using the v component? 2. You propose a "formal" definition of energy in eq. 1, but this is not actually the energy that you define in eq. 2. Consider using different notations, in order to avoid confusion.*

Response: To 1: We changed the definition of the wind vector to "where $v$ is the meridional wind".
To 2:
Reviewer: *l. 95: this dry component is not the dry static energy (DSE), or is it? It should not include a kinetic energy term;*

Response: Indeed, the dry-static energy from the first version of the manuscript also includes the kinetic energy, strictly it is the dry energy without "static". The kinetic component is some orders of magnitude smaller than the other two (Peixoto and Oort, 1992), hence both are essentially similar. However, for being precise we remove the "static" in the manuscript.

Reviewer: *ll. 111-116: it is clear that because of cylindrical symmetry, cross terms in eq. 6 and 7 cancel, but this should be stated explicitly;*

Response: We introduced a sentence: "Note, that the cross terms $a_n^E$ and $a_m^v$ with $n \neq m$, and similarly $b_n^E$ and $b_m^v$, vanish since the Fourier components feature an orthogonal basis."

Reviewer: *ll. 124-125: the choice of the mentioned wavelengths for scale separation shall be rather commented here than in Sect. 3;*

Response: The discussion of the chosen wavelengths is dedicated the entire Section 3, so it would be to long to insert it here. The method is in general independent from the chosen wavelengths, we only mention it to improve the interpretability for the reader.

Reviewer: *ll. 139-141: I am surprised that the most basic constraint to the width of the synoptic-scale eddies, i.e. the Rossby deformation radius, is not mentioned;*

Response: We do now mention it: "The theoretical scale (wavelength) of baroclinic eddies is given by 3.9 times the Rossby deformation radius, and hence estimated to be 4,000 km by (Vallis, 2017, p.354) and 4,800 km by Stoll et al. (2021)."

Reviewer: *ll. 168-169: this finding clearly suggests that eddies below this scale possess a dispersion relation (cfr. Dell'Aquila et al. 2005) and this is in line with expectation about baroclinic eddies in mid-latitudes. I wonder if a space-time decomposition could be provided in order to show this relation;*

Response: These are interesting thoughts. We include a comparison of our results to that study: "The spectral decomposition of the annual-mean energy transport, $\widehat{vE}$, at different latitudes reveals that most eddies smaller than 8000 km are of transient nature, whereas most of the quasi-stationary transport is at scales larger than 8000 km (Fig. 2). This is in good agreement with Dell'Aquila et al. (2005) finding in the average of the extra-tropical NH, that most standing eddies occur at zonal wavenumbers 3 - 5 and that propagating eddies at wavenumber 3 feature a typical time period of around a months whereas small eddies, here associated to the synoptic scale, are characterised by weekly and daily periods." However, a space-time decomposition is outside the scope of the current study.

Reviewer: *ll. 189-190: is it something new? Wasn't it already found in other works on the topic of wavenumber vs. traditional transient/quasi-stationary decomposition?*

Response: This finding may not be completely new, however, we are not aware of studies comparing the traditional with the scale separation of atmospheric energy transport. If the reviewer is aware of a study that compares the different composition methods, we would gladly refer to it here.
We outline the problematic that both are sometimes considered similar in the Introduction: "... quasi-stationary transport is often associated with the planetary scale, which appears to imply that planetary transport is irrelevant in the SH (e.g. Trenberth and Stepaniak, 2003a). In contrast, transport by transient eddies is often associated with baroclinic eddies at the synoptic scale (e.g. Trenberth and Stepaniak, 2003a). However, transport at other scales could be of transient character as well."

Reviewer: *l. 222: what does "seamless" mean in this context?*

Response: We mention our interpretation of "seamless" a few sentences before, which is a term utilised in the cited study: "The annual-mean, zonal-mean poleward energy transport, vE, for both hemispheres (black lines in Fig. 4a) as noted by Trenberth and Stepaniak (2003b)."

Reviewer: *ll. 225-226: is this "analytical form" of the transport reflecting any physical mechanism?*

Response: We remove the analytical form which is somewhat difficult to recognise in the perspective of plotting both hemispheres together as done in Figure 5. Indeed, the maximum and minimum in the poleward moisture transport are caused by different mechanisms leading to the moisture transport, which is long known and which we explain in the remainder of the section.

Reviewer: *l. 271: I wonder if it could be possible to comment on the absence of a (even weak) polar cell in the NH;*

Response: We add two sentences on the topic: "Different to previous studies by for example Peixoto and Oort (1992) a NH polar cell is not evident in the here-utilised ERA-5 dataset, neither in the annual-mean nor in the summer or winter season (Sec. 4.2). In the Arctic, energy transport is dominated by eddies, whereas zonal symmetric katabatic flows, as observed in the Antarctic, do not develop due to the lack of a large ice dome centred over the pole."

Reviewer: *ll. 280-281: if the mesoscale component is negligible, why would you need to include it in the synoptic transport?*

Response: We add the last part of the sentence to the manuscript to answer the questions: "Due to its negligible role, we include the mesoscale into the synoptic transport for the remainder of this study, such that the sum of all components yield the total transport"

Reviewer: *l. 291: this seems to suggest symmetry in the location of the ITCZ, whereas we know that the ITCZ is located about 8N in the annual mean;*

Response: Studies like the below cited indeed indicate approximately a symmetry in the ITCZ. However, we changed the text a bit: "The location separating northward and southward total transport, the energy flux equator (Adam et al., 2016), is at around 10°latitude in the summer hemisphere (Fig. 6d). This is linked to the zonal-mean ITCZ, associated with the ascending branch of the Hadley circulation (Adam et al., 2016)." Surely their are different definitions of the zonal-mean ITCZ so it is a bit challenging. If the reviewer disagrees with our interpretation, we would be glad if she or he could point towards some literature.

Reviewer: *ll. 295-296: is it something new, or was it already seen by performing more naive scale separations in the past?*

Response: Thanks for the remark. Indeed it could be recognised in the wavenumber-based scale separation as well. Hence, we add a subsentence: "... in broad agreement with results from wavenumber separated transport of the NH mid-latitudes in Lembo et al. (2019) ..."

Reviewer: *l. 312: given that you are discussing some hypotheses here, I think it*

*makes sense to expand a little bit on this, rather than barely referring to a subsequent paper;*

Response: We expand a bit with a short argumentation: "A subsequent study points towards that the annual-mean energy transport is induced by the meridional energy gradient in the manner of a diffusion process with a globally almost constant diffusion coefficient, hence larger transport in one component reduces the temperature gradient, leading to less transport in another component. Differently, moisture is a tracer of the atmospheric circulation and therefor not described by a diffusion process such that the components do not compensate in a similar manner as for the energy transport. "

**Technical corrections**

Reviewer: *l. 20: replace "hereby" with "thereby";*

Response: Thanks

Reviewer: *l. 66: authors could be more specific on the choice of the variables. Replace "temperature" with "air temperature" and "humidity" with "specific humidity" (?);*

Response: Done as advised.

Reviewer: *Figure 2: in the caption dashed lines shall be also defined, together with solid lines;*

Response: The dashed lines are replaced by solid lines. Further we replaced: "The wavenumbers corresponding to some wavelengths are presented by black curves. The solid curves at 2,000 and 8,000 km denote the separation between meso, synoptic and planetary scale."
by: "The wavenumbers corresponding to wavelengths of 2000, 4000, 6000 and 8000 km are depicted by black curves. The first and last of the curves separate the transport into the meso, synoptic and planetary scale."

Reviewer: *l. 341: "astonishing" does not seem the right term in this context. Consider changing it (maybe "surprising", "remarkable"?);*

Response: Thanks for the suggestion, we replaced it by remarkable and slightly changed the formulation of the sentence after suggestion of reviewer 2.

Reviewer: *l. 345: replace "mechanism" with "mechanisms";*

Response: Thanks.

Reviewer: *l. 353: remove the first "of" and comma before "to";*

Response: Thanks.

**Response to reviewer 2.**

We thank the referee for the thorough and constructive review. The comments are contributing to an overall improved manuscript. Note that some formulations in the new manuscript are slightly different than here since they were changed again in a proofreading process.

Reviewer: *In general, I think the presented arguments and discussion in this manuscript could be a bit clearer. A stronger focus could also be on the dynamical understanding of the contribution of the different spatial or temporal scales. I think this is important, as the goal of the paper is to use the somewhat new wavelength based consideration to get a better understanding of the atmospheric circulation. Maybe the authors could also make it clearer what exactly is the new contribution of their study, because this wavelength vs wavenumber consideration was also discussed in previous studies. It is not fully clear to me, if it is the application to the energy transport analysis, the more in detail and systematic approach of the comparison with more traditional scale separation or if the authors see this procedure as something fully new.*

Response: We attempt to be more clear on the main new contributions of our study. We therefor reformulate some of our arguments. For example at the end of the second paragraph we changed the formulation "Here we revise the traditional separation and compare it to a partition based on the spatial scale."
to: "In this study, we combine the traditional separation of the meridional energy transport with a revised partition by spatial scales to improve our understanding of the manner the atmosphere transports energy polewards."
To connect our decomposition with the underlying physical mechanism, we add a dynamical argument for our separation captured by a new Figure 3: The here-defined synoptic energy transport at scales between 2000 - 8000 km is associated with enhanced meridional temperature gradients a few days before, hence appears to be of baroclinic origin, whereas the planetary transport appears to be little influenced by the meridional temperature gradient and some planetary waves are stronger when the temperature gradient was reduced a few days before.

Reviewer: *In terms of getting a deeper understanding, I would also suggest to give more context and insight into the sensitivity test, e.g. the sensitivity on the choice of the wavelength scale is only done for the climatological annual mean signal as well as for the impact of the quasi-stationary contribution to the synoptic scale, but the results are then analysed for different seasons without discussing how those time and spatial scale choices might impact those results.*

Response: Generally, the new Figure 3 provides more evidence to separate at 8000 km in order to separate between baroclinically-induced synoptic transport and planetary transport originating from other mechanisms. Hence, the sensitivity analysis is not that important. However, we provide a sensitivity analysis in Figure 1 that separates the annual-mean and seasonal-mean transport between the synoptic and

(a) $\widetilde{vE}$; annual; 6000 km

(b) $\widetilde{vE}$; annual; 8000 km

(c) $\widetilde{vE}$; annual; 10000 km

(d) $\widetilde{vE}$; winter; 6000km

(e) $\widetilde{vE}$; winter; 8000km

(f) $\widetilde{vE}$; winter; 10000km

(g) $\widetilde{vE}$; summer; 6000km

(h) $\widetilde{vE}$; summer; 8000km

(i) $\widetilde{vE}$; summer; 10000km

Figure 1: (b) As Figure 5 of the manuscript the annual-mean energy transport by different components, but including the quasi-stationary component of the synoptic transport. (a) and (c) as (b) but separating between the synoptic and planetary transport at a wavelength of 6000 and 10000 km, respectively. (d-f) as (a-c), but for the winter-mean transport and (g-i) for summer-mean transport.

planetary transport at the wavelength of 6000, 8000 and 10000 km. It also includes individually the quasi-stationary contribution of the synoptic transport, that is rather small in all panels and hence for simplicity not depicted as an own component in the manuscript. Note, however, that the quasi-stationary contribution of the synoptic transport is part of the synoptic transport.

Comparing the separations at different wavelengths: Clearly the strength of synoptic and planetary transport is dependent on the separation wavelength. For example, the synoptic transport is much stronger when separated at 10000 km than at 6000 km and consequently the planetary transport smaller. This is not surprising since the band of waves with wavelengths between 6000 and 10000 km transports a considerable amount of energy, as can be seen in Fig. 2. However more importantly, the form of the curves in synoptic and planetary transport is little effected by the separation wavelengths, hence the characteristics of the transport components does not depend on the threshold. Also the different transport patterns in summer and winter, where synoptic transport is approximately equally important in the former, whereas planetary transport becomes relevant in the winter, is not effected by the threshold. Hence, we are confident that our results are quite robust independent of the exact chosen separation wavelength.

Reviewer: *In particular the neglect of the quasi-stationary synoptic signal should be discussed in more detail, as especially for different seasons one might expect a shift of the contributions from the planetary to the synoptic scale or the quasi-stationary to the transient scale, dependent on how the threshold was chosen.*

Response: This is a misunderstanding, since the quasi-stationary synoptic signal is not neglected in the study. It is part of the synoptic signal, just not investigated individually.

Reviewer: *I have the feeling the reader might not get a much deeper understanding from the authors introduced wavelength based consideration, as those mentioned aspects are not really explained sufficiently and the reduced sensitivity testing might leave the reader with several open questions. E.g. in the conclusions the choice of the wavelength threshold is said to be based mainly on the intuition of the authors, which is not very convincing. Without a clearer sensitivity testing (which should also be discussed in the conclusions) it might be hard to convince the reader of this approach.*

Response: We provide more evidence in Figure 3 that the separation is successful to capture baroclinic versus non-baroclinic transport. We expand our discussion in Section 3 and in the Conclusions as we point out later in the specific comments.

Reviewer: *So in general, I think, highlighting the different approaches is already very useful, so therefore this manuscript is already quite useful. But I think the authors should slightly improve in the presentation of their results and put more effort into highlighting what deeper understanding the reader can get from this approach. This*

*also includes a more convincing sensitivity study to better understand the impact of the threshold choices on the differences between the scale contributions, not only for the annual mean signal but also the different seasons, because that is when I would expect the largest impact (e.g. dominant wavenumbers in midlatitudes are quite different in summer and winter). In the following I included further specific comments.*

Response: We present an extended sensitivity study for different seasons in Figure 1.

**Specific comments**

Reviewer: *line 30: "fast-varying" means everything faster than a month here? As the authors highlight different disturbances, such as polar lows, maybe faster varying would be a better description as also much slower disturbances are part of the same "fast-varying" group.*

Response: We agree and adapt the formulation "faster-varying".

Reviewer: *lines 38-41: This sentence somehow seems to suggest that the previous studies were missing an important point, i.e. the latitude dependent spatial scale of a wavenumber. However, this might not be that relevant for their studies, e.g. Rothlisberger et al. (2019) are interested in the occurrence of wave patterns in the midlatitudes. As they are not primarily interested in the meridional transport of energy and how the wave separation changes with latitude, this seems to be another question and is therefore not necessarily a disadvantage of their method. I would suggest that the authors make this point clearer, i.e. using a fixed wavenumber range might be fine for studies that look at specifics dynamics at a fixed latitude range, but it could be misleading if one does investigate the dynamics across a large range of latitudes. It might be worth mentioning that similar studies of wave patterns in the midlatitudes, based on wavenumbers, did account for this by varying the wavenumber depending on the latitude, e.g. Wolf and Wirth (2017), Diagnosing the horizontal propagation of Rossby wave packets along the midlatitude waveguide, see their Fig. 6). So using a method based on wavenumbers, does not necessarily mean that one cannot account for this effect.*

Response: We agree and thank for the reference. We rewrote the following formulation: "These studies separate the transport by a wavenumber which is independent of the latitude as depicted in Figure 1b. However, the wavelength associated with a given wavenumber is latitude dependent (Fig. 2). Therefore the partitioning by wavenumber, for example between wave 3 and 4 as performed in many of the previously mentioned studies, leads to convergence of all eddy transport to the planetary scale towards the poles, whereas synoptic transport may be overestimated at low latitudes (Fig. 1b). Wave 4 for instance corresponds to a wavelength of 8200 km at 35, but only to 2600 km at 75, which can be interpreted to represent different spatial scales. Accordingly, Heiskanen et al. (2020) recommend to consider the threshold for separation between two wavenumbers with care."

to: "These studies separate the transport by a wavenumber which can be associated with a wavelength (spatial scale) for a given latitude. For investigation of the scale of the transport across a specific latitude (or a small zonal band) a fixed wavenumber for separation is appropriate. However, the wavelength associated with a given wavenumber is latitude dependent which needs to be accounted for when defining the wavenumber separating spatial scales (Heiskanen et al., 2020). Wave 4 for instance corresponds to a wavelength of 8200 km at 35°, but only to 2600 km at 75°, which can be interpreted to represent different spatial scales.

So far the energy transport across all latitudes has only been separated by a fixed wavenumber as shown in Figure 1b and presented by (Graversen and Burtu, 2016). Such a partitioning by wavenumber, for example between wave 3 and 4 as performed in many of the previously mentioned studies, has two caveats. 1) Towards the poles all eddy transport convergence to the planetary scale (Fig. 1b). 2) In the subtropics, wavenumbers 1 - 3 appear to miss parts of planetary transport, as can be inferred from quasi-stationary eddies in the subtropical SH (Fig. 1a), being considerably larger than the planetary transport captured by wavenumbers 1 - 3 (Fig. 1b)."

We refer to the study of Wolf and Wirth (2017) in terms of a latitude-dependent separation by wavelength in the method section. However, this was not applied on the energy transport.

Reviewer: *line 54: Referring to: "However, transport at other scales could be of transient character as well." and "In this study, we are pointing out that the separation..." I would suggest the authors be more specific here and more clearly about the point they try to make or slightly reformulate this paragraph. This paragraph seems to suggest that this point (other scales can be of transient character, too) is an important new point tackled in this study. However, there are studies which explicitly highlight the point that quasi-stationary waves can be transient, as it is already part of their name (quasi-stationary, so not necessarily stationary). So I think the authors should highlight their real contribution better, namely doing this more systematically. Although many other studies also highlight this point, or modify their wavenumber based method to account for the latitude dependent spatial scale effect, this study is systematically investigating this issue in more detail.*

Response: In the here-applied "traditional" decomposition by Oort and Peixóto (1983), the energy transport by monthly-mean eddies is referred to as quasi-stationary, whereas transport anomalies from the monthly mean are transient. This is formulated in L29 - 30 (old version): "Traditionally, the eddy transport is separated into a quasi-stationary and a transient component (Fig. 1a), with the former representing monthly-mean eddies and the latter faster-varying deviations from this mean (Oort and Peixóto, 1983)." Following the definition of the traditional decomposition, quasi-stationary eddies may vary from month to month, are however strictly distinct from transient eddies.

Here, the new contribution is the wavelength-based decomposition which we compare to the traditional decomposition. We argue that the new wavelength-based decomposition is more generally applicable to all latitudes than the wavenumberbased decomposition.

We would be interested to know which studies are with "Although many other studies also highlight this point, or modify their wavenumber based method to account for the latitude dependent spatial scale effect".

Reviewer: *line 61: I would suggest to mention all following sections here, not only the data section.*

Response: We added a short overview of the manuscript: "These questions are targeted in Sections 3 and 4. The main results are then summarised and discussed in Section 5. However, first the utilised data and methods are presented."

Reviewer: *line 71: Maybe "In this study, we take a zonal-mean perspective of the local atmospheric energy transport, ..." to make it right away clearer to the reader that this is a local approach.*

Response: Good suggestion. We rearranged the paragraph and changed the sentence: "Hence, to compare the local importance of the atmospheric energy transport across all latitudes, we take a zonal-mean perspective which provides the transport through an atmospheric column with one metre width. "

Reviewer: *lines 71-74: Maybe the authors can be more specific here about the differences of the two approaches, as from this description I don't really see the difference between a zonal mean or the zonal integral. Maybe I am missing an important point, but not fully obvious to me why I should expect the peaks at different latitudes.*

Response: The differences is simple, the zonal integral is the zonal mean multiplied by the longitude circle. We try to make this more clear by changing the formulation from: "In this study, we take a zonal-mean perspective of the atmospheric energy transport, which provides the transport through an atmospheric column with one metre width. Hereby, it provides a local measure of the transport, and differs from other studies that zonally integrate the transport along each longitude circle (Graversen and Burtu, 2016; Peixoto and Oort, 1992; Trenberth and Caron, 2001). However, the computed zonal integral of the energy transport from ERA5 (Fig. S1a) confirms the transport in these studies. For instance, the zonal-integrated poleward transport peaks at $4.8{\times}10^{15}$ W in the NH and $5.6{\times}10^{15}$ W in the SH at 41° latitude in both hemispheres. The latitude of maximum zonal-mean transport is slightly higher at 45°(Fig. S1b). Further, the average transport in the polar regions is more easily assessed by the zonal-mean transport as it is not influenced by converging latitudes."

To: "The zonal integral of the energy transport from ERA5 (Fig. S1a) confirms the transport in found in previous studies (Graversen and Burtu, 2016; Peixoto and Oort, 1992; Trenberth and Caron, 2001). For instance, the zonal-integrated poleward transport peaks at $4.8{\times}10^{15}$ W in the NH and $5.6{\times}10^{15}$ W in the SH at 41° latitude in both hemispheres. By computing the zonal integral of the energy

transport, which depends on the length of the longitude circle, the transport becomes small at high latitudes since the longitudes converge (Fig. S1a). However, the local transport, expressed by the zonal mean, is considerable also in the polar regions (Fig. S1b). Hence, to compare the local importance of the atmospheric energy transport across all latitudes, we take a zonal-mean perspective which provides the transport through an atmospheric column with one metre width. Hereby, for example the latitude of maximum zonal-mean transport is at 45° latitude (Fig. S1b)."

Reviewer: *line 76: ylabel: length instead of lenth*

Response: Thanks for spotting mistake. We changed the y-labels to "Mean transport".

Reviewer: *line 81: Why is this extensive smoothing necessary, so why not only before the calculation of derivatives (to get rid of possible large unrealistic gradients) but also afterwards again?*

Response: Indeed the smoothing is not really necessary and we remove everywhere beside before the computation of the derivatives for the convergence to reduce the noise.

Reviewer: *lines 86-87: Is this formulation ("only possible from a time-mean perspective") justified? In general the quasi-stationary transport does not need to be based on monthly mean fields, so if this would be adapted then it would also be possible on smaller temporal scales. Further, the authors mention that other studies are "normally" basing this on monthly fields, which suggests that not all are following this procedure, which would not support the "only possible from" formulation.*

Response: To solve the mentioned issues and to be more precise, we change the formulation from: "This comparison is only possible from a time-mean perspective, since the quasi-stationary transport is normally derived based on monthly-mean fields (Oort and Peixóto, 1983)."
To: "This comparison is only possible from a time-mean perspective, since the computation of quasi-stationary eddy transport requires a predefined time period over which the eddies are considered stationary, which is traditionally set to be one month (Oort and Peixóto, 1983)."

Reviewer: *line 91: Only v appears in this equation, not v, so I would suggest not to refer in the notation explanation to the full wind.*

Response: We change the notion.

Reviewer: *line 93: Maybe the authors could be more explicit by what they mean when they say "in three ways that can be applied in combination". Those are three different ways to identify energy transport, so what do the authors have in mind if they talk about combining those separations?*

Response: We agree that the formulation was not very clear and hence change it: "We decompose the total atmospheric energy transport, $\widetilde{vE}$, in three ways that can be applied in succession:"
In the following, we reformulate that section in order to explain how the decomposition is performed in succession.

Reviewer: *line 94: Replace "archived" by "achieved".*

Response: Thanks for spotting mistake.

Reviewer: *line 104: Replace reference to "Fig. 1a" by "Fig. 1".*

Response: It is intended to refer to Figure 1a which displays the traditional decomposition.

Reviewer: *lines 117-118: Equation (7) represents the zonal mean for one particular time step, correct? Figure 2 however shows a temporal average, so I would suggest to explicitly mention this in lines 117-118 so that the reader does not get confused with the notation ([·] only representing zonal mean).*

Response: That is correct. We hence add some explanation to the method section: "This partition is applied to the instantaneous co-variability, $vE$ in Equation 1, resulting in the wave-separated total transport, as well as to the transport by monthly-mean fields, $\overline{vE}$, comprising term 2 and 3 of Equation 3, resulting in the wave-separated quasi-stationary transport, $\widetilde{vE}^{q-s}$. The annual-mean transport by each quasi-stationary wave ($[\overline{vE}]_n$ in Eq. 7) as function of latitude is displayed in Figure 2a). The wave-separated transient transport, $\widetilde{vE}^{tran}$, (Fig. 2b) is derived by subtracting the quasi-stationary from the total transport, computed from the instantaneous co-variability. "

Reviewer: *line 126: Continuous separation shown in Fig. S2b, not S2a.*
Response: Indeed, we changed the order in the manuscript.

Reviewer: *lines 144-145: I find it difficult to follow the conclusion for the use of a band between 2000-8000 km. The reference threshold is about 4000-4700 km, which would correspond to a wavenumber of 6-7 at about 45° latitude. According to the authors there is some variability associated with the scale of synoptic cyclones and in the fourier decomposition also neighbouring wavenumbers contribute strongly to the energy transport. But 2000 km (8000 km) at 45°N would be represented by a wavelength between 14 and 15 (3 and 4) and those larger scales could also be associated already with stationary or quasi-stationary longitudinally extended waves.*

Response: We show in Figure 2 that little transport of eddies at scales smaller than 8000km is of quasi-stationary character.

Reviewer: *Those wave patterns can still be associated with the smaller than plane-tary wave scales and therefore be considered as part of the smaller synoptic band, but the authors explicitly highlight the link to the much smaller synoptic cyclones. To me it is not clear from this paragraph what the authors want to have included in their synoptic band. Also from the sentence with the reference to the synoptic Rossby waves (lines 145-146) it is not clear if the authors want to have those in-cluded or if they just tolerate this to be able to capture most of the energy transport with the synoptic scales they are interested in. I suggest the authors to rephrase this paragraph to make this clearer. If the authors are indeed only interested in the smaller scales as synoptic cyclones, I think there is some more justification neces-sary for the choice of the upper threshold of 8000 km, because I would expect no synoptic cyclone at 45°N would be represented by a wavenumber 3 to 4.*

Response: For clarification: At 45° our separation assigns wavenumber 3 and more than half of wavenumber 4 to the planetary scale.

As suggested, we rephrase the paragraph: "The synoptic scale is supposed to include most energy transport associated with eddies developing by baroclinic instability (Holton and Hakim, 2013; Vallis, 2017). The synoptic eddies are perceived as cy-clones and anticyclones in the sea-level pressure that are interacting vertically with an upper-tropospheric oscillation of the jet stream, often associated with transient synoptic Rossby waves (e.g. Ali et al., 2021; Röthlisberger et al., 2019). The the-oretical scale (wavelength) of baroclinic eddies is given by 3.9 times the Rossby deformation radius, $L_d$, and hence estimated to be $4000\,\mathrm{km}$ by (Vallis, 2017, p.354) and $4,800\,\mathrm{km}$ by Stoll et al. (2021). Note, that a low (high) pressure system spans half a wavelength, and has accordingly a typical size of around $2000\,\mathrm{km}$.

A wavelength band between $2000 - 8000\,\mathrm{km}$ appears appropriate to capture the ma-jority of the transport associated with baroclinically-induced synoptic eddies for two reasons: (i) Synoptic eddies, such as cyclones and anticyclones, feature some variability in their size, but with a typical diameter between 1000 and $4000\,\mathrm{km}$. (ii) The non-local Fourier decomposition of the energy transport in situations of localised synoptic cyclones captures considerable amount at neighbouring waves to the cyclone (Heiskanen et al., 2020, Fig. 3)"

Further, we add another analysis that demonstrates the chosen threshold at 8000km to separate between baroclinically-induced energy transport and transport created differently.

Reviewer: *lines 198-199: Why is this contribution (quasi-stationary component of synoptic transport) not further investigated? The reason seems to be that it its contribution to the synoptic scale is rather small (although up to 30%) and doesn't really fit into the category of quasi-stationary planetary scale? Excluding this con-tribution seem to suggest that the introduced categories of synoptic and planetary or transient and quasi-stationary have some difficulties capturing the processes they are supposed to capture. This part therefore could also be seen as some measure of category uncertainty, excluding it fully seems a bit surprising.*

Response: The quasi-stationary component of synoptic transport is still included in the synoptic transport, just not investigated separately since it is small (see Fig. 4 and Fig. 1). We think our formulation and reasoning was a bit unclear and reformulated the paragraph. It now includes a reason for not further investigating the quasi-stationary synoptic transport. Our method of scale separation is different from the "traditional separation", if both would agree 100%, our method would be redundant. So, we do not share the interpretation that this is "category uncertainty".

Old paragraph: "..., the synoptic transport is mainly (70 - 100%) of transient nature at all latitudes, which coincides with the transient character of synoptic cyclones and Rossby waves of short wave length. Hence in the following, the quasi-stationary component of synoptic transport is not further investigated. "

New paragraph: "..., the synoptic transport is mainly (70 - 100%) of transient nature at all latitudes, which coincides with the transient character of synoptic cyclones and Rossby waves of short wave length. The small quasi-stationary contribution (0 - 30%) to the synoptic transport is attributed to preferred spatial locations for synoptic activity. For instance, the NH Atlantic sector features more cyclonic activity than other longitudes, which in the time-mean reveals as increased quasi-stationary transport. This can be inferred as Rydsaa et al. (2021) show a large time-mean synoptic transport in the Atlantic sector for strong latent transport events in winter at 70° N. However, in a zonal-mean perspective the quasi-stationary contribution to the synoptic transport is small (< 30%) compared to its transient part, and hence for the sake of simplicity the synoptic transport is not separated into a transient and quasi-stationary contribution in the remainder of this study.

Reviewer: *lines 204-205: Maybe the authors should specify here a bit more to which main results they refer, as the lines for different wavelengths are very different, e.g. in terms of the contribution from synoptic and planetary scales in mid-latitudes (synoptic much stronger for 10000km, but weaker for 6000km). So the wavelength has a huge impact on the separation between planetary and synoptic scales (qualitative different conclusions). Therefore I think the authors should give some more context here, for what results/analysis this separation does not matter.*

Response: The supplement includes a short discussion of the difference mentioned by the reviewer and the similarities that we refer to in the manuscript:

Supplement (slightly rewritten): "In order to test the sensitivity of the scale separated energy transport for different values of the wavelength used for separation, the latter is varied (Fig. S6). Clearly, more (less) transport is associated with the synoptic scale when separation wavelength is increased (decreased). This is simply a result from the wavelength band between 6,000 and 10,000 km comprising a considerable amount of the energy transport (see also Fig. S3). Hence, the strength of the synoptic as compared to the planetary component is influenced by varying the separation wavelength. However, the important features of planetary and synoptic waves are similar, such as the maximum in the synoptic transport around 45°latitude, the maximum of the planetary around 60°latitude, almost symmetrical

structures in both hemispheres, and similar seasonal behaviour (not shown)."

In the manuscript, we add a short reference to the supplement by the last part of the following sentence: "The main results of this study are not affected by the exact choice of the separation wavelength which is shortly discussed in the Supplement."

Reviewer: *lines 211-213: I would suggest to avoid the reference to Fig. S6, as this figure is about the impact of using different scales. Further Fig. 4 is the relevant figure, which shows the signal for both hemispheres, so there does not seem to be any need to additionally refer to another figure. Further, is a particular reason that only the x-axis for panel d to f is scaled, but not for panel a to c? I found it initially a bit confusing when I tried to compare transport and convergence fields.*

Response: First part: We agree that it is better omit the reference to Figure S6. Second part: We understand that the different x-axis scales can confuse the comparison of the panels. The reason is that in a to c, we attempt to show the transport at each latitude which we consider best visible by a linear x-axis. In d to f, we scale the axis "such that the integrated convergence in each component becomes zero" (which we add to the legend of the figure). This way it appears more intuitive that the energy in each component is redistributed. We would be interested if you have a opinion on how to best combine these competing considerations.

Reviewer: *line 216: I would exclude "almost", because the curves are similar in the sense that they have the shape. I guess almost refers to the amplitude difference, but this is explained in the following.*

Response: We agree and removed "almost".

Reviewer: *lines 225-226: Do the authors really mean an inverse sine function in Fig. 4b? I find it hard to identify this curve behaviour in this plot. Further isn't there a difference between NH and SH (next sentence in lines 226-227 seems to suggest this is not the case)?*

Response: Indeed the inverse sine function is a bit difficult to see when both hemispheres are plotted together. Hence, we reformulated the paragraph and now also discuss the differences between the hemispheres.

Old: "To a first order, the annual-mean moisture transport, $\widetilde{vQ}$, resembles the inverse of a sine curve in each hemisphere with an exponentially decaying tail towards the poles (Fig. 5b). Hence, moisture transport in the tropics is equatorward and poleward in the subtropics and extra-tropics with a maximum around 40° latitude. This leads to moisture divergence in the non-equatorial tropics and sub-tropics and convergence in the equatorial regions and extra-tropics (Fig. 5e)."

New: "The total annual-mean moisture transport, $\widetilde{vQ}$, of both hemispheres features equatorward extremes at around 10°(Fig. 5b), a poleward maxima at around

40°and decaying tails towards the poles. This leads to moisture divergence in the non-equatorial tropics and sub-tropics, whereas moisture convergences in the equatorial regions and extra-tropics (Fig. 5e). The moisture transport is mainly stronger in the Southern than Northern Hemisphere, likely due to more evaporation on water surfaces of the Southern Hemisphere. Further, some moisture is transported from the SH across the equator leading to the highest convergence of moisture at around 7° N, which is in the annual mean the approximate location of the intertropical convergence zone (ITCZ)."

Reviewer: *line 229: Maybe referring to the curves as showing a plateau is a bit too much, at least for both hemispheres. Maybe this can be rephrased slightly with saying "more plateau-like" or something similar.*

Response: As suggested, we reformulate "features a plateau" to "is plateau-like".

Reviewer: *lines 249-250: But isn't that what Fig. 4a is showing, that plan q-s is much stronger in NH than SH? This statement refers to previous interpretation of this manuscript or other studies (Trenberth and Stepaniak, 2003)? I don't really understand the contradiction, because the statement seems to agree with the figure. If the contradiction refers to the similar curves of the planetary signal for both hemisphere, then I also don't understand the contradiction, because individual parts of this signal (q-s and transient) do not necessarily need to have the same behaviour. I would suggest to rephrase this paragraph to make this clearer.*

Response: We try to make our point more clear and reformulate the paragraph. Old: "The planetary energy transport is almost similar in both hemispheres, different from the quasi-stationary transport that is mainly relevant in the NH (Fig. 5a). The planetary transport is similar in the subtropics and low mid-latitudes and only approximately 20% weaker in the higher mid-latitudes of the SH than the NH. This is in contrast to the previous interpretation that planetary transport being represented by the quasi-stationary component is mainly relevant in the NH (e.g. Trenberth and Stepaniak, 2003). This is the case since the planetary transport has a highly relevant transient component in the SH (Fig. 4)."

New: "The planetary energy transport is similar in both hemispheres, different from quasi-stationary transport which is mainly relevant in the NH (Fig. 1a, 5a). The latter is in agreement with Trenberth and Stepaniak (2003) pointing that quasi-stationary transport is a primary factor in the extratropical NH. They associate this quasi-stationary transport to the planetary scale, which they do not prove but which is confirmed by this study (Fig. 4). A new finding, that could partly be inferred from Fig. S3 of Lembo et al. (2019), is the almost symmetry of the planetary energy transport in both hemispheres. This symmetry could not been anticipated by the consideration of quasi-stationary transport since the planetary transport in the SH is mainly of transient character (Fig. 4), in agreement with Mo (1986).
In both hemispheres, the planetary transport is similar in the subtropics and low mid-latitudes and only approximately 20% weaker in the higher mid-latitudes and

polar region of the SH than the NH. Hence, eddies at similar spatial scales are transporting the energy in both hemispheres (see also Fig. 2), which is likely due to similar physical mechanisms in both hemispheres forming the energy-transporting eddies."

Reviewer: *lines 260-262: Why is this exactly most remarkable, or as this agrees with other studies do the authors have an explanation? As the strength of the wave guides in the NH and SH are very different, with the SH having a stronger jet, a separation by an identical time filter for identical wavenumber signals would lead on one hemisphere (SH) to the identification of a transient signal whereas on the other hemisphere (NH) as a quasi-stationary signal.*

Response: We agree with the reviewer and our formulation was a bit imprecise. Hence, we changed from: "The most remarkable difference between the hemispheres is that planetary waves are transient in the extra-tropical SH, whereas often quasi-stationary in its northern counterpart, which agrees with Peixoto and Oort (1992)."

To: "These planetary waves are mainly transient (Fig. 4: 70%) in the mid-latitudinal SH, whereas more often (60%) quasi-stationary in its northern counterpart, which agrees with Peixoto and Oort (1992). However, in the high latitudes of the SH, considerable amount of the planetary transport is quasi-stationary."

Reviewer: *lines 295-296: This is linked to a previous point, referring to lines 198-199. During winter there is a stronger wave guide and if considered in a power spectra spaned by wavenumber and latitude, more power at all latitudes is shifted towards smaller wavenumbers (compared to summer). This means, during summer the center of the power distribution will be located higher wavenumbers. It is therefore possible that the contribution of quasi-stationary signal is included in the synoptic scale. The authors mentioned in lines 198-199 that this part (quasi-stationary signal within synoptic scale) will not be considered as it does not represent such a large contribution. However, in summer this contribution could be larger.*

Response: See Fig. 1, the quasi-stationary contribution to the synoptic transport is small in summer.

Reviewer: *As the authors investigate all season (annual mean) and all seasons individually, I think they should be more specific about this point, e.g. when discussing this contribution in lines 198-199 they should already consider the seasonal differences. If this contribution would be larger in summer, their argument of not considering this contribution because of their small contribution seems more problematic. Further, the authors discuss the seasonal differences with Fig. 5 while excluding this part completely (synoptic and quasi-stationary). I would find it very interesting to see this contribution also included in those plots as it also shows the sensitivity of the analysis to the defined classes (synoptic, planetary, q-s, etc) and differences in the dynamics for the different seasons.*

Response: It appears that most of this is due to a misunderstanding that we hope is partly resolved by the response to lines 198-199. The synoptic transport includes both a transient and quasi-stationary component, just the separation is not presented.

Reviewer: *lines 296-297: As mentioned in the previous point, it could also be that synoptic signals dominates in summer because the q-s is no longer mainly part of the planetary signal, but during summer part of the synoptic signal (but not considered).*

Response: As mentioned earlier, the quasi-stationary synoptic transport is included in the synoptic transport, just not presented individually (in the manuscript. However, it is shown in Fig. 1. Since it is small in all seasons, it is not considered to be of major interest.

Reviewer: *This somehow is strongly linked to the important point of this paper, highlighting the point that defining patterns on wavenumbers can be problematic because of the latitude dependence. The authors show convincingly the relevance of this point in great detail. But isn't it also relevant to consider the timescale of the wave patterns as function of the season, as it was for the spatial scale as function of latitude?*

Response: It would be indeed relevant to investigate the timescale of planetary and synoptic transport events. As work of a follow-up study, we find that planetary events feature a mean lifetime of around a week (somewhat shorter in the SH), whereas synoptic events last for around 3 days.

Reviewer: *lines 309-310: What do the authors mean by "are rather summing up to the total variance"? The signal is not summing up, but that is also not really expected that the variances sum up, as already stated by the authors in the previous sentence. The variability fraction also shows that the individual parts show stronger variability than the total signal, so there is the same kind of compensation between the different signals with overall smaller values. I would suggest the authors specify more in detail why this panel is so much different to the other one to better support their statement and following hypothesis.*

Response: We rewrite, from: "In contrast to the total transport, the variability of the moisture transport components, $\widetilde{vQ}$, are rather summing up to the total variance (Fig. 7b)."
To: "In contrast to the total energy transport, the variability of the total moisture transport, $\widetilde{vQ}$, is larger than the variability of its individual scale components (Fig. 7b). Hence, the moisture transport components are not compensating each other in the same manner as the total energy transport components. Instead, the compensation of the components is in form of the dry energy (Fig. 7c)."

Reviewer: *lines 313-315: This statement is explicitly about the Meri-part? If so, the authors should make this clear, because the statement in its general form does*

*not seem to be supported by Fig. 6.*

Response: We add a sentence for clarification: "The large total variability in the moist and dry energy transport is almost entirely due to variability in the meridional components, which is not surprising since the meridional components are responsible for most of the moist and dry energy transport in the tropics (Fig. 5b,c, 6b,c,e,f)."

Reviewer: *line 318: Is approximately 10% correct? It seems that all values in the extratropics exceed 10%, with values up to about 20%.*

Response: It is the orange line in Figure 7a. We rewrite the sentence: "In the extratropics, the planetary transport, $\widetilde{vE}_{plan}$, exhibits the largest inter-annual variability and varies by approximately 10% in the mid-latitudes and 15-20% in the polar regions (Fig. 7). "

Reviewer: *lines 320-323: This is again linked to my comments about lines 198-199 and 295-296. The planetary variability is strongly linked to the variability of the q-s component. If the q-s signal is linked to the strength and/or location of the wave guide, isn't it possible that part of it fall into the synoptic part for specific years? This would then be visible in the synoptic q-s part, but this is not part of this analysis here.*

Response: As explained before, the q-s synoptic part is included in the synoptic part.

Reviewer: *line 332: I think it is not a really strong and convincing argument to base the choice of length scales on the intuitive understanding. For example, with my intuitive understanding I would have chosen a slightly different range of length scales. I understand that any choice will always be subjective, because there is no truth for doing the separation, but it can be stated like this or also refered to similar length scales in other studies, I would however suggest to not base the argument on intuition.*

Response: We expand the discussion on this topic: "We demonstrate that a separation between synoptic and planetary eddies at a wavelength of 8000 km is physically useful since it distinguishes between waves preceded by enhanced and reduced meridional temperature gradients. Hence, that synoptic eddies at wavelengths smaller than 8000 km, are mainly baroclinically induced, whereas different physical mechanisms are at work for larger eddies. The same wavelength is also in approximate agreement with the traditional separation between transient and quasi-stationary eddies, as most wave transport at wavelengths smaller than 8000 km is of transient character, whereas most quasi-stationary transport occurs at the planetary scale larger than 8000 km. Despite the latter, considerable planetary energy transport is of transient character, especially in the extratropical SH.
The separation between synoptic and planetary eddies at a wavelength of 8000 km

appears large. However, most baroclinically-induced and transient energy transport organises at a wavelength around 5000 km at all latitudes (Fig. 2, 3), well in agreement with the predicted length by dry-baroclinic theory (Vallis, 2017). However, the baroclinically-induced and transient energy transport occurs in a wavelength band approximately between 2000 and 8000 km, hence separating at around 5000 km would be misleading. It should further be noted that one synoptic wave includes both a low and a high pressure systems, hence that synoptic cyclones and anticyclones feature a typical diameter of between 1000 and 4000 km, or that the typical distance between two independent (anti)cyclones is between 2000 and 8000 km. This appears appropriate from comparison with weather maps."

Reviewer: *lines 333-336: This spatial separation is such an important feature of the presented analysis. Therefore, I think the authors should include a comment here in the conclusion about the sensitivity. I included a statement in the result section about this sensitivity as well, which might be relevant here as well. I would include this sensitivity test (Fig. S6) even in the main manuscript and discuss the identified differences. If there are no relevant differences then I would agree to keep it in the supplementary material and just say that the results are not sensitive to the exact choice of length scale. However, as included in my previous comment about this issue, I think there are relevant differences. If the authors agree on this point, I think it makes sense to include it here in the main manuscript, if the authors disagree, then I think they should make it clearer what relevant part of the results are similar for the different length scales.*

Response: See the second response that presents the Figure 1.

Reviewer: *line 342: Referring to "rather narrow band". Is this really a narrow band? This range represents a wavenumber range of about 3.5 to 14.2 at 45°latitude, which doesn't seem very narrow.*

Response: Indeed, the narrow appears misplaced and we removed it. Hence the sentence was changed from: "It is astonishing that despite all possible eddies, waves at scales in the rather narrow band between 2,000 and 8,000 km are responsible for the majority of the meridional energy transport for the whole extra-tropics."
To: "It is remarkable that in the large range of atmospheric eddies, those at scales in the band between 2,000 and 8,000 km are responsible for the majority of the meridional energy transport for the whole extra-tropics."

Reviewer: *line 347: I think it should be stated somewhere in the conclusion that the q-s part of the synoptic scales is excluded. I would further suggest to include a whole paragraph to discuss this exclusion, why it was done and what possible impacts could be for the results or the sensitivity of the study. How relevant is this excluded part for the different seasons?*

Response: As stated earlier, this is a misunderstanding.

**References**

[revised manuscript text omitted]

---

## Referee Report (RR1)

**General comment**

The authors did a very thorough job in answering all my raised points and they modified their manuscript according to those suggestions, giving more explanations where I thought the original manuscript could be a bit clearer and they improved on highlighting their contributions to the research, which I find did help to improve the manuscript. Therefore I do not really have many comments left. Small minor points are given below under the "specific comments" section.

**Specific comments**

**lines 49-52:**
I would combine this paragraph with the following, as here the authors introduce the consideration of the meridional length scales with their short conclusion that a fourier transform along longitude circles does not make much sense. I find this paragraph a bit confusing, why introducing this new aspect when directly concluding that it does not make sense? At least in the next paragraph the authors present their explanations why investigating the meridional scale is not really necessary as it is linked to the zonal scale. I would suggest to combine this into one paragraph and maybe introducing it that the authors focus on the zonal scale of the investigated features. After this they can give their reasoning why further looking explicitly at the meridional scale is not necessary and hence, specify that in the following spatial scale always refers to the associated zonal scale. If considering the meridional scale individually is not necessary, then I would also exclude the part about the fourier transform along longitude circles as it does not contribute to their arguments.

**line 116:**
I would suggest to replace "normal" circulation by another description, because it is not clear what normal really means. The authors talk about a temporal averaged circulation to get rid of the large noisy day to day variability, so maybe it could be refered to as temporal averaged or averaged instead of normal.

**lines 117-119:**
This refers to one of my previous comments in the previous revision process, not seeing quasi-stationary signals necessarily based on time-mean fields. The authors refer to the monthly mean averages as the traditional method to identify the quasi-stationary signal and they also include a reference. So I guess this is the justification to refer to this part as the quasi-stationary signal and use this terminology with the explained meaning in their manuscript.

However, a quasi-stationay signal could also be identified for example by temporal low pass filter, excluding the faster transients and only keeping the slowly moving signals as the quasi-stationary part, there is no real reason why one should not do it like this. Depending on the exact separation between quasi-stationary and faster transients for

the time filtering, a monthly mean field and the associated monthly averaged temporal filtered field of the wave amplitudes could be different, because it is a temporal mean of a signal that is not necessarily stationary. Due to this, I am not sure I agree with the author's description that "This comparison is only possible from a time-mean perspective... quasi-stationay eddy transport requires a predefined time period over which the eddies are considered stationary". This conclusion is based on their definition of a quasi-stationary signal, which is based on using a time-mean, but from this perspective it is by definition "only possible from a time-mean perspective", but that is a consequence of their method. So the presented results can only be shown based on a time-mean perspective (due to the definition of the method), but it is not necessarily true that investigating the quasi-stationary signal is only possible based on a time-mean perspective.

**line 162:**
reference to the equation number not working (??).

**line 256:**
wavelength instead of wavenumber

---

## Author Response (AR2)

**Response to editor**

*The authors have addressed most of the reviewers' concerns from the first round, with fairly substantial revisions made to the manuscript. Both reviewers are satisfied, and point out only some small remaining issues. I encourage the authors to look after these final points, in particular the comments of reviewer 2 about clarifying the writing and the fact that "quasi-stationary" can be defined in more than one way.*

Response: We thank the editor for the overall positive response and we respond to the reviewers as recommended. We include a short mention that quasi-stationary eddies can be defined differently.

*In addition, I have three notes:*

*1) In the caption of Fig. 2, I don't understand this statement: "At each latitude the wave of maximal poleward energy transport is denoted in grey, where values are masked if the wave is responsible for less than 5% of the transport at the latitude with maximum transport." The "where" suggests that the masked values are associated with the grey markings, but I don't see any masking. Or are these two separate statements?*

Response: Maybe the word mask is not accurate and we understand that it is communicated a bit unclear. The "where" should refer to the grey markings: If look at Fig. 2b you see that the markings are missing in the tropics.
To improve the formulation we change "At each latitude the wave of maximal poleward energy transport is denoted in grey, where values are masked if the wave is responsible for less than 5% of the transport at the latitude with maximum transport."
To: "At each latitude the wave of maximal poleward energy transport is displayed with a grey dot. The dot is not displayed at latitudes where this wave is responsible for less than 5% of the total quasi-stationary/ transient transport of the latitude with maximum transport."

*2) Reviewer 1 has noted some remaining typos/grammatical errors but there are also quite a few others throughout the manuscript. In some cases, the errors interfere with the meaning (e.g., L311-312: "In the SH subtropics..." I'm assuming the "which" clause is not meant to refer to the NH, as currently written, but rather the (larger) transport in the SH?) I recommend that the authors do a thorough proofreading pass.*

Response: Indeed that passage was a unclear formulated. We rephrased it to the following: "However, in the subtropic and mid-latitudes the energy transport is ∼15% larger in the SH than the NH, which is balanced by more oceanic transport in the NH (Trenberth and Caron, 2001)."
We also do a thorough proofreading as the editor suggests.

*3) I would tend to use "partitioning" rather than "partition" whenever you are talking about the way in which the energy transport is separated. I believe it is the more commonly used terminology, and also sounds more natural to me. However, I leave this up to the authors.*

Response: We agree and replace "partition" by "partitioning".

**References**

Trenberth, K. E. and Caron, J. M.: Estimates of meridional atmosphere and ocean heat transports, Journal of Climate, 14, 3433–3443, 2001.

**Response to reviewer 1**

We thank the reviewer for the positive feedback and for spotting multiple typos.

*l. 16: remove the first "is";*

Response: Done as adviced.

*l. 115: remove "Swanson and Pierrehumbert, 1997";*

Response: Thanks.

*l. 162: Missing reference to Equation;*

Response: Thanks.

*l. 195: replace "to" with "with";*

Response: Thanks.

*l. 312: replace "in the than in the the NH" with "than in the NH";*

Response: Thanks.

*l. 349: replace "to" with "with";*

Response: Thanks.

*l. 444: replace "Consequentially" with "Consequently";*

Response: Thanks.

*l. 464: insert "that" between "such" and "the";*

Response: Thanks.

*l. 491: replace "as has long been" with "as long";*

Response: Thanks.

*l. 499: replace "Also" with "Other";*

Response: Thanks.

**Response to reviewer 2**

We thank the reviewer for coming with some interesting thoughts and comments that further improved the manuscript.

*lines 49-52: I would combine this paragraph with the following, as here the authors introduce the consideration of the meridional length scales with their short conclusion that a fourier transform along longitude circles does not make much sense. I find this paragraph a bit confusing, why introducing this new aspect when directly concluding that it does not make sense? At least in the next paragraph the authors present their explanations why investigating the meridional scale is not really necessary as it is linked to the zonal scale. I would suggest to combine this into one paragraph and maybe introducing it that the authors focus on the zonal scale of the investigated features. After this they can give their reasoning why further looking explicitly at the meridional scale is not necessary and hence, specify that in the following spatial scale always refers to the associated zonal scale. If considering the meridional scale individually is not necessary, then I would also exclude the part about the fourier transform along longitude circles as it does not contribute to their arguments.*

Response: We agree that these two paragraph are a bit confusing and too specific for an introduction. Therefore, as advised, we combine the paragraphs, considerably shorten them and move them to the end of the method section. It now reads as:
"Generally, the scale separation based on the Fourier decomposition is non-local, implying that the whole latitude circle influences the obtained eddies (Heiskanen et al., 2020). Therefore the Fourier decomposition is useful if the transport across the circle is governed by similar eddy scales, which we can observe from meteorological weather maps along zonal bands. Arguably, the zonal and meridional scales of atmospheric eddies match: from the investigation of meteorological weather maps, we know (i) that synoptic-scale cyclones are to a first order circular, and (ii) that the meridional extent, i.e. the amplitude, of Rossby waves, appears to roughly match the distance between a trough and a ridge in the zonal direction. Hence, as many other studies (Graversen and Burtu, 2016; Lembo et al., 2019, e.g.), we interpret the zonal wavenumber of an eddy, which is associated with a zonal wavelength at a given latitude, as its spatial scale."

*line 116: I would suggest to replace "normal" circulation by another description, because it is not clear what normal really means. The authors talk about a temporal averaged circulation to get rid of the large noisy day to day variability, so maybe it could be refered to as temporal averaged or averaged instead of normal.*

Response: We follow the suggestion of the reviewer and replace "However, here we focus on the annual-mean and season-mean energy transport, firstly to investigate the "normal" circulation, and secondly to compare the traditional separation of the transport by quasi-stationary and transient eddies with a separation based on

spatial scales."

with: "However, here we focus on the annual-mean and season-mean energy transport, firstly to investigate the time-mean behaviour of the atmospheric circulation, and secondly to compare the traditional separation of the transport by quasi-stationary and transient eddies with a separation based on spatial scales. "

*lines 117-119: This refers to one of my previous comments in the previous revision process, not seeing quasi-stationary signals necessarily based on time-mean fields. The authors refer to the monthly mean averages as the traditional method to identify the quasi-stationary signal and they also include a reference. So I guess this is the justification to refer to this part as the quasi-stationary signal and use this terminology with the explained meaning in their manuscript. However, a quasi-stationay signal could also be identified for example by temporal low pass filter, excluding the faster transients and only keeping the slowly moving signals as the quasi-stationary part, there is no real reason why one should not do it like this. Depending on the exact separation between quasi-stationary and faster transients for the time filtering, a monthly mean field and the associated monthly averaged temporal filtered field of the wave amplitudes could be different, because it is a temporal mean of a signal that is not necessarily stationary. Due to this, I am not sure I agree with the author's description that "This comparison is only possible from a time-mean perspective... quasi-stationay eddy transport requires a predefined time period over which the eddies are considered stationary". This conclusion is based on their definition of a quasi-stationary signal, which is based on using a time-mean, but from this perspective it is by definition "only possible from a time-mean perspective", but that is a consequence of their method. So the presented results can only be shown based on a time-mean perspective (due to the definition of the method), but it is not necessarily true that investigating the quasi-stationary signal is only possible based on a time-mean perspective.*

Response: Here we refer to the quasi-stationary definition utilised by Peixoto and Oort (1992) and Trenberth and Stepaniak (2003a). We agree that quasi-stationary transport can be defined in a different manner as the mentioned temporal low pass filter. Indeed, defining the quasi-stationary transport in such a manner, one would not need to perform the comparison based on temporal-mean fields. Hence, we rewrote the introductory paragraph to Section 2.3:

"The atmospheric energy transport and its components are characterised by a large day-to-day variability (Messori and Czaja, 2013; Swanson and Pierrehumbert, 1997). However, here we focus on the annual-mean and season-mean energy transport, firstly to investigate the time-mean behaviour of the atmospheric circulation, and secondly to compare the newly-introduced separation of the eddy transport based on spatial scales – described below – with the conventional separation into quasi-stationary and transient transport introduced by Oort and Peixóto (1983) and commonly used in the literature (Kaspi and Schneider, 2013; Trenberth and Stepaniak, 2003a,b, e.g.). The conventional separation obtains quasi-stationary eddies from monthly-mean fields; hence a comparison is only possible from a time-mean perspective. Quasi-stationary eddies could be defined differently, for example, by

application of a temporal low pass filter, such the methods could be compared without taking time averages. However, this would prevent a direct comparison with the literature and require a new ad-hoc analysis, and is beyond the scope of this study."

*line 162: reference to the equation number not working (??).*

Response: Thanks for spotting the mistake. We changed some formulations in the subsection and the references to the equations were checked.
*line 256: wavelength instead of wavenumber*

Response: Thanks for spotting the mistake.

**References**

Graversen, R. G. and Burtu, M.: Arctic amplification enhanced by latent energy transport of atmospheric planetary waves, Quarterly Journal of the Royal Meteorological Society, 142, 2046–2054, 2016.

Heiskanen, T., Graversen, R. G., Rydsaa, J. H., and Isachsen, P. E.: Comparing wavelet and Fourier perspectives on the decomposition of meridional energy transport into synoptic and planetary components, Quarterly Journal of the Royal Meteorological Society, 146, 2717–2730, 2020.

Kaspi, Y. and Schneider, T.: The role of stationary eddies in shaping midlatitude storm tracks, Journal of the atmospheric sciences, 70, 2596–2613, 2013.

Lembo, V., Messori, G., Graversen, R., and Lucarini, V.: Spectral Decomposition and Extremes of Atmospheric Meridional Energy Transport in the Northern Hemisphere Midlatitudes, Geophysical Research Letters, 46, 7602–7613, https://doi.org/https://doi.org/10.1029/2019GL082105, 2019.

Messori, G. and Czaja, A.: On the sporadic nature of meridional heat transport by transient eddies, Quarterly Journal of the Royal Meteorological Society, 139, 999–1008, 2013.

Oort, A. H. and Peixóto, J. P.: Global angular momentum and energy balance requirements from observations, in: Advances in Geophysics, vol. 25, pp. 355–490, Elsevier, 1983.

Peixoto, J. P. and Oort, A. H.: Physics of climate, American Institute of Physics, 1992.

Swanson, K. L. and Pierrehumbert, R. T.: Lower-tropospheric heat transport in the Pacific storm track, Journal of the atmospheric sciences, 54, 1533–1543, 1997.

Trenberth, K. E. and Stepaniak, D. P.: Covariability of components of poleward atmospheric energy transports on seasonal and interannual timescales, Journal of climate, 16, 3691–3705, 2003a.

Trenberth, K. E. and Stepaniak, D. P.: Seamless poleward atmospheric energy transports and implications for the Hadley circulation, Journal of Climate, 16, 3706–3722, 2003b.